# Transport or Discard: Robust Unbalanced Optimal Transport for Cross-Domain Policy Adaptation

**Wenyu Chen** [1] [*]  **Yujia Zhang** [1] [*]  **Wei Guo** [1]  **Linli Ma** [1]  **Yanbo Wang** [1]  **Pinle Qin** [1]  **Jianchao Zeng** [1]

## Abstract

Cross-domain offline reinforcement learning leverages a source dataset to improve policy learning in a data-scarce target domain, but dynamics mismatch makes many source transitions kinematically infeasible and can cause negative transfer. Recent non-parametric geometric methods (e.g., standard optimal transport and k-nearest neighbors) avoid overfitting yet often yield only relative rankings under an implicit matching or retrieval budget, making performance sensitive to hand-tuned thresholds when the true cross-domain overlap is unknown. We formulate availability estimation as *soft subset selection* by learning a source reweighting that geometrically aligns with the target. We propose **R**obust **O**ffline unbalanced **O**ptimal **T**ransport (ROOT): (i) a robust ambiguity set for uncertainty under limited target samples, and (ii) an unbalanced transport objective that penalizes mass deviation, enabling a principled *transport-or-discard* mechanism. ROOT thus down-weights or discards high-cost source samples rather than forcing them onto the target support. Moreover, the induced weights decay exponentially with transport cost, guaranteeing outlier suppression. On D4RL dynamics-shift benchmarks, ROOT improves downstream offline RL and outperforms strong baselines on most tasks without task-specific threshold tuning.

## 1. Introduction

Offline Reinforcement Learning (RL) has emerged as a promising paradigm for learning policies directly from static datasets (Levine et al., 2020; Zhang et al., 2026a). However, the performance of offline policies is fundamentally limited by the coverage and quality of the available data. To mitigate data scarcity, cross-domain offline RL (Liu et al., 2022) seeks to leverage source-domain datasets to improve policy learning in a data-limited target domain. A central challenge is dynamics mismatch: many source transitions can be dynamically incompatible with the target environment, and naively incorporating them often induces severe negative transfer.

Early data filtering methods usually estimate cross-domain relevance with parametric models, e.g., domain classifiers for dynamics-gap prediction (Liu et al., 2022) or mutual-information-based filtering (Wen et al., 2024). However, under limited target data settings, such high-capacity estimators can overfit and yield unreliable importance weights. This has motivated non-parametric geometric alternatives, such as Optimal Transport (OT) (Chapel et al., 2020) and $k$-Nearest Neighbors (KNN) (Kramer, 2013), which measure cross-domain proximity without training auxiliary discriminators (Lyu et al., 2025; Le Pham Van et al., 2025).

While effective in avoiding overfitting, these approaches still produce relevance estimates under an implicit matching or retrieval budget. As a result, they provide relative relevance rankings rather than absolute availability scores. Specifically, standard mass-preserving OT and fixed-$k$ KNN retrieval implicitly enforce a transfer budget, i.e., a predetermined amount of source data that must be matched or retained, irrespective of actual cross-domain overlap. If a fixed threshold retains 50% of source data while the true transferable overlap is only 20%, many irrelevant transitions are inevitably imported; conversely, if the true overlap is 80%, the same threshold discards valuable data. This exposes a core mismatch: existing transfer rules prescribe *how much to transfer*, while true cross-domain compatibility determines *how much is transferable*. This raises a key question:

*Can we design a principled mechanism that moves beyond fixed transfer quotas and instead computes sample-wise availability scores that adapt to unknown dynamics mismatch?*

We answer this question by casting availability estimation as a *soft subset selection* problem: learning a reweighted source distribution that minimizes its geometric discrepancy

[*]Equal contribution  [1]North University of China, Taiyuan, China. Correspondence to: Jianchao Zeng <zjc@nuc.edu.cn>.

*Proceedings of the 43$^{rd}$ International Conference on Machine Learning*, Seoul, South Korea. PMLR 306, 2026. Copyright 2026 by the author(s).

from the target distribution. However, directly using balanced Wasserstein matching is ill-suited because the target distribution is only observed through limited samples and, under dynamics mismatch, not all source mass should be transported. We therefore propose ROOT, a Robust Offline unbalanced Optimal Transport framework, which (i) introduces a robust ambiguity set to model target-side uncertainty from scarce target data, and (ii) adopts an unbalanced transport objective that relaxes source mass conservation by penalizing marginal deviation, yielding a principled transport-or-discard mechanism. ROOT automatically down-weights or discards high-cost source samples instead of forcing them onto the target support; we further show that the induced relevance weights decay exponentially with transport cost, providing a theoretical guarantee for outlier suppression. On D4RL benchmarks (Fu et al., 2020) with diverse dynamics shifts, ROOT outperforms strong cross-domain baselines on most tasks without task-specific threshold tuning. Ablation studies further indicate robustness to the relaxation hyperparameters.

## 2. Preliminaries

### 2.1. Offline Reinforcement Learning

A reinforcement learning problem can be modeled as a Markov decision process (MDP) $\mathcal{M} = (\mathcal{S}, \mathcal{A}, P, R, \gamma)$, where $\mathcal{S}$ and $\mathcal{A}$ denote the state and action spaces, $P(s'|s, a)$ is the transition dynamics, $R(s, a)$ is the reward function, and $\gamma \in [0, 1)$ is the discount factor (Sutton & Barto, 1998; Zhang et al., 2024a;b; 2025). The goal is to learn a policy $\pi(a|s)$ maximizing the expected discounted return $J(\pi) = \mathbb{E}_{\pi, P}\left[\sum_{t=0}^{\infty} \gamma^t r_t\right]$. In offline RL, interaction with the environment is prohibited; the agent learns policies solely from a fixed dataset $\mathcal{D} = \{(s_k, a_k, r_k, s'_k)\}_{k=1}^K$.

### 2.2. Cross-Domain Offline Reinforcement Learning

Following existing literature on cross-domain offline RL (Zhang et al., 2026b; Qiao et al., 2026a), we consider a source domain $\mathcal{M}_{\text{src}}$ and a target domain $\mathcal{M}_{\text{tar}}$, which share the same state-action space and reward function but differ in transition dynamics: $P_{\text{src}}(s'|s, a) \neq P_{\text{tar}}(s'|s, a)$. The agent has access to two offline datasets: a limited target dataset $\mathcal{D}_{\text{tar}}$ collected in the target domain, and a typically larger source dataset $\mathcal{D}_{\text{src}}$ collected in the source domain, which may contain transitions that are dynamically incompatible or kinematically infeasible under the target dynamics. Our goal is to leverage $\mathcal{D}_{\text{mix}} = \mathcal{D}_{\text{tar}} \cup \mathcal{D}_{\text{src}}$ to learn a policy $\pi$ that maximizes the expected discounted return in the target domain.

### 2.3. Wasserstein Distance and Optimal Transport

The Wasserstein distance serves as a geometry-aware metric for quantifying the discrepancy between two probability distributions (Cuturi, 2013), which incorporates the geometry of the underlying sample space. Formally, given two empirical probability measures $\mu \in \mathbb{R}_+^M$ and $\nu \in \mathbb{R}_+^N$ with $\sum_i \mu_i = \sum_j \nu_j = 1$, the Wasserstein distance is defined via the Kantorovich formulation of OT:

$$\mathcal{W}_c(\mu, \nu) = \min_{\Gamma \in \Pi(\mu, \nu)} \langle \Gamma, C \rangle = \min_{\Gamma \in \Pi(\mu, \nu)} \sum_{i=1}^M \sum_{j=1}^N \Gamma_{ij} C_{ij}, \tag{1}$$

where $C \in \mathbb{R}_+^{M \times N}$ is the cost matrix with entries $C_{ij} = c(x_i, y_j)$ representing the geometric distance between samples. The optimization is constrained over the set of admissible couplings $\Pi(\mu, \nu)$, defined as:

$$\Pi(\mu, \nu) = \{\Gamma \in \mathbb{R}_+^{M \times N} \mid \Gamma \mathbf{1}_N = \mu, \, \Gamma^\top \mathbf{1}_M = \nu\}. \tag{2}$$

In the standard formulation, these marginal constraints enforce strict mass conservation: the row and column marginals of the transport plan $\Gamma$ must exactly match the source and target distributions, respectively. Consequently, all probability mass in $\mu$ is forced to be transported to $\nu$ while minimizing the total transport cost. This mass-conservation property is central to balanced OT, but it can become restrictive when only a subset of source samples is compatible with the target domain.

## 3. Methodology

In this section, we present the ROOT framework for cross-domain offline RL. In Section 3.1, we first formulate source selection as a soft subset selection problem, aiming to learn a reweighted source distribution that is geometrically close to the target distribution. We then show that this formulation leads to an idealized transport problem and identify two key obstacles: the target distribution is unknown under limited samples, and enforcing a fixed amount of transported source mass can be harmful under partial domain overlap. In Sections 3.2 and 3.3, we address these obstacles through target-side uncertainty modeling and source-side unbalanced relaxation, respectively. We further provide a theoretical guarantee showing that ROOT suppresses high-cost source outliers. The overall framework is illustrated in Figure 1.

### 3.1. Problem Formulation

To estimate a sample-wise availability score for each source transition, we first formulate source selection as a soft subset selection problem, defined as follows:

**Definition 3.1** (Soft Subset Selection). Let $\hat{\nu}_{\text{src}}$ denote the empirical source distribution, and let $\nu_{\text{tar}}^*$ denote the

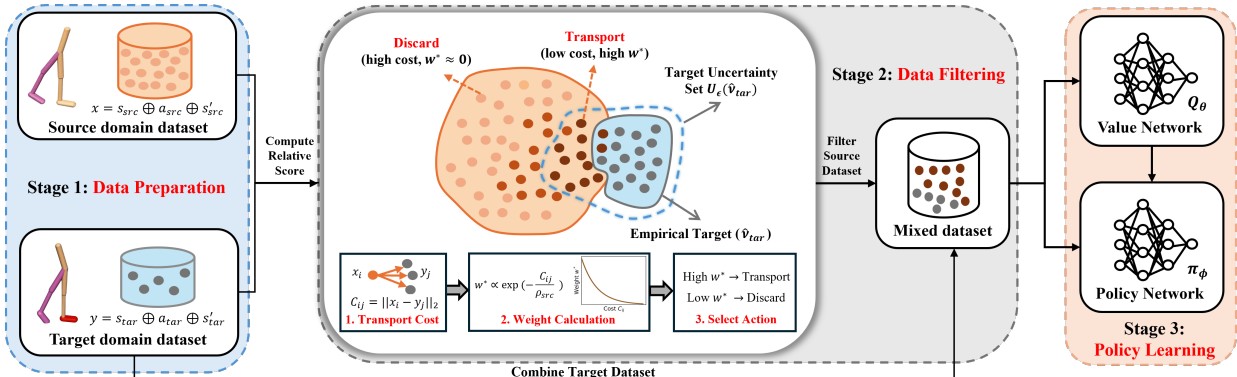

*Figure 1.* **Overview of the ROOT framework.** Our method consists of three phases: **(a) Data preparation phase**, which constructs joint transition tuples $x$ and $y$ from source and target datasets. **(b) Data filtering phase**, which solves an unbalanced optimal transport problem and assigns high relevance weights to source samples that are geometrically consistent with the target, while exponentially suppressing incompatible ones. **(c) Policy learning phase**, which uses the mixed dataset to train the target policy.

unknown true target distribution over the joint transition space $\mathcal{X}$. We seek a nonnegative weighting function $w : \mathcal{X} \to \mathbb{R}_+$ that reweights the source distribution as $\nu_w(x) := w(x)\hat{\nu}_{\text{src}}(x)$ so that the reweighted source distribution is geometrically close to the target distribution. The optimal source reweighting is defined by:

$$\min_{w \geq 0} \ \mathcal{W}_c(\nu_w, \nu_{\text{tar}}^*) \quad \text{s.t.} \quad \mathbb{E}_{x \sim \hat{\nu}_{\text{src}}}[w(x)] = 1. \quad (3)$$

Here, $\mathcal{W}_c$ denotes the Wasserstein distance induced by the ground cost metric $c(x, y)$. We define each transition tuple as $x = s_{\text{src}} \oplus a_{\text{src}} \oplus s'_{\text{src}}$ for the source domain and $y = s_{\text{tar}} \oplus a_{\text{tar}} \oplus s'_{\text{tar}}$ for the target domain, so that $c(\cdot, \cdot)$ measures transition-level compatibility.

This formulation searches for a valid probability distribution supported on the source dataset that best matches the target geometry. We use the Wasserstein distance because it provides a finite and interpretable discrepancy measure even when some source samples lie outside the target support, which is essential under partial cross-domain overlap (Panaretos & Zemel, 2019).

To make the objective in Definition 3.1 amenable to optimization, we reformulate the Wasserstein term via its Kantorovich formulation (Chizat et al., 2018), which turns the problem into a minimization over a transport plan $\Gamma$:

$$\min_{\Gamma \geq 0} \ \langle \Gamma, C \rangle \quad \text{s.t.} \quad \Gamma^\top \mathbf{1} = \nu_{\text{tar}}^*, \quad \sum_{i,j} \Gamma_{ij} = 1. \quad (4)$$

Here, $C$ denotes the empirical cost matrix induced by the ground metric $c$, with entries $C_{ij} = c(x_i, y_j)$. A detailed derivation is provided in Appendix B.1. Given the optimal plan $\Gamma^*$, the source availability scores can naturally be recovered from the source marginal of $\Gamma^*$, i.e., $\nu_w = \Gamma^* \mathbf{1}$.

Although Eq. (4) is convex, solving it directly faces two fundamental obstacles:

- **Unknown Target Distribution.** The true target distribution $\nu_{\text{tar}}^*$ in the marginal constraint is inaccessible. In practice, we only observe a finite empirical approximation $\hat{\nu}_{\text{tar}}$, which may be sparse and noisy under limited target data.

- **Fixed Transport Quota.** The total-mass constraint $\sum_{i,j} \Gamma_{ij} = 1$ enforces a fixed amount of source mass to be transported to the target. Under partial domain overlap, this creates a quota effect: suppressing an outlier requires redistributing its mass to other source samples, even when the remaining source samples may also be dynamically incompatible. Such forced mass allocation can inflate the influence of mismatched transitions and amplify negative transfer.

To overcome these challenges, ROOT introduces two relaxation mechanisms in the following sections: a target-side uncertainty relaxation for limited target coverage, and a source-side unbalanced relaxation that enables incompatible source mass to be down-weighted or discarded.

## 3.2. Target-Side Uncertainty under Limited Coverage

The first challenge in solving Eq. (4) stems from the inaccessibility of the true target distribution. Simply replacing it with the empirical estimate $\hat{\nu}_{\text{tar}}$ forces the transport plan to match a finite, potentially sparse sample set. This leads to severe overfitting, where the algorithm forces mass onto specific empirical points rather than the underlying target manifold.

To mitigate this, we introduce a KL ambiguity set around the empirical target marginal. Instead of forcing the transported target marginal $\nu = \Gamma^\top \mathbf{1}$ to exactly match $\hat{\nu}_{\text{tar}}$, we allow it to vary within a controlled neighborhood:

$$\mathcal{U}_\delta(\hat{\nu}_{\text{tar}}) = \{\nu \in \Delta_N \mid D_{\text{KL}}(\nu \| \hat{\nu}_{\text{tar}}) \leq \delta\}, \quad (5)$$

where $\Delta_N$ denotes the probability simplex over the $N$ empirical target samples. This gives the following robust target-matching problem:

$$\min_{\Gamma \geq 0, \nu \in \mathcal{U}_\delta(\hat{\nu}_{\mathrm{tar}})} \langle \Gamma, C \rangle \text{ s.t. } \Gamma^\top \mathbf{1} = \nu, \quad \sum_{i,j} \Gamma_{ij} = 1. \quad (6)$$

A detailed derivation is provided in Appendix B.2. The above formulation replaces exact matching to a sparse empirical marginal with robust matching to a set of plausible target marginals. As a result, the transported target marginal can adapt to the geometry and density of the observed target samples, while remaining close to the empirical target distribution.

By applying Lagrangian relaxation to the KL constraint, we incorporate the target uncertainty set into the objective. Under standard regularity conditions, for a given radius $\delta$, there exists a corresponding multiplier $\lambda_{\mathrm{tar}} > 0$ such that Eq. (6) can be written in the following penalized form:

$$\min_{\Gamma \geq 0} \langle \Gamma, C \rangle + \lambda_{\mathrm{tar}} D_{\mathrm{KL}}(\Gamma^\top \mathbf{1} \| \hat{\nu}_{\mathrm{tar}}) \text{ s.t. } \sum_{i,j} \Gamma_{ij} = 1. \quad (7)$$

Here, $\lambda_{\mathrm{tar}}$ controls how strongly the transported target marginal is tied to the empirical target marginal. A larger $\lambda_{\mathrm{tar}}$ enforces closer agreement with $\hat{\nu}_{\mathrm{tar}}$, while a smaller value permits stronger adaptation of the target marginal under limited coverage. In our experiments, we use a fixed $\lambda_{\mathrm{tar}}$ across tasks and show in Section 5 that ROOT is stable over a broad range of values.

### 3.3. Source-Side Relaxation for Transport-or-Discard

The second challenge comes from the fixed total-mass requirement in Eq. (7). Although source reweighting allows different source samples to receive different weights, enforcing $\sum_{i,j} \Gamma_{ij} = 1$ still requires a fixed amount of source mass to be transported to the target. Under partial domain overlap, this creates a quota effect: reducing the mass assigned to one incompatible source transition necessarily redistributes that mass to other source samples. As a result, the learned weights may reflect a forced allocation budget rather than the true usability of each source transition.

To remove this fixed-quota coupling, we relax the source side from a probability distribution to a nonnegative measure. Instead of requiring all source mass to be transported, we allow the transported source marginal $\Gamma\mathbf{1}$ to deviate from the empirical source distribution $\hat{\nu}_{\mathrm{src}}$, and penalize this deviation through a source-side generalized KL term:

$$\mathcal{R}_{\mathrm{src}}(\Gamma) = \lambda_{\mathrm{src}} D_{\mathrm{KL}}(\Gamma\mathbf{1} \| \hat{\nu}_{\mathrm{src}}). \quad (8)$$

This relaxation allows the optimizer to pay a finite marginal penalty for discarding high-cost source mass, rather than forcibly transporting it to the target support.

After removing the fixed total-mass constraint, both marginal penalties are interpreted as generalized KL divergences between nonnegative measures. Combining the source-side relaxation with the target-side uncertainty relaxation yields the final ROOT objective:

$$\min_{\Gamma \geq 0} \langle \Gamma, C \rangle + \lambda_{\mathrm{tar}} D_{\mathrm{KL}}(\Gamma^\top \mathbf{1} \| \hat{\nu}_{\mathrm{tar}}) + \lambda_{\mathrm{src}} D_{\mathrm{KL}}(\Gamma\mathbf{1} \| \hat{\nu}_{\mathrm{src}}). \quad (9)$$

Compared with balanced OT, Eq. (9) no longer enforces a fixed total transported mass. Therefore, source samples are not selected according to a predetermined retention ratio; instead, their relevance is determined jointly by transport geometry and marginal relaxation penalties.

After solving Eq. (9), we obtain the optimal transport plan $\Gamma^*$. Let $\rho_i^* = (\Gamma^*\mathbf{1})_i$ denote the transported source marginal for source sample $x_i$. The corresponding sample-wise relevance weight is recovered as

$$w_i^* = \frac{\rho_i^*}{\hat{\nu}_{\mathrm{src},i}} = M \sum_{j=1}^N \Gamma_{ij}^*, \quad i = 1, \ldots, M, \quad (10)$$

where the second equality holds for a uniform empirical source distribution $\hat{\nu}_{\mathrm{src},i} = 1/M$. Detailed derivations are provided in Appendix B.3. The following proposition shows that these weights decay exponentially with geometric mismatch, which formalizes the transport-or-discard mechanism.

**Proposition 3.2** (Exponential Suppression of Irrelevant Transitions). *Let $w_i^*$ be the relevance weight for source sample $x_i$ obtained from the optimal coupling of the regularized ROOT objective, and let $C_{\min}(x_i) = \min_j C(x_i, y_j)$ be its minimum geometric cost to the target support. Then, there exists a target-dependent constant $B_{\mathrm{tar}} > 0$ such that*

$$e^{-\frac{C_{\min}(x_i) + B_{\mathrm{tar}}}{\lambda_{\mathrm{src}}}} \leq w_i^* \leq e^{-\frac{C_{\min}(x_i) - B_{\mathrm{tar}}}{\lambda_{\mathrm{src}}}}.$$

See Appendix C for detailed derivation.

**Remark:** Proposition 3.2 formalizes ROOT's selectivity: the relative weight assigned to $x_i$ decays (up to a target-dependent offset $B_{\mathrm{tar}}$) exponentially with its minimum transport cost $C_{\min}(x_i)$. The hyperparameter $\lambda_{\mathrm{src}}$ acts as a temperature controlling the sharpness of this decay—smaller $\lambda_{\mathrm{src}}$ yields more aggressive suppression of high-cost transitions. This property explains why ROOT can down-weight dynamics-incompatible source transitions without relying on hand-tuned per-task filtering thresholds.

**Summary.** ROOT provides a principled and efficient framework for computing absolute availability scores for each source transition. By relaxing both marginal constraints of standard OT via KL penalties, it transforms the original soft selection formulation into a tractable unbalanced OT problem. The resulting weights $w^*$ are theoretically

---

**Algorithm 1** Block-wise Generalized Sinkhorn

---

1: **Input:** Source data $\mathcal{D}_{\mathrm{src}}$, Target data $\mathcal{D}_{\mathrm{tar}}$, Hyperparameters $\lambda_{\mathrm{src}}, \lambda_{\mathrm{tar}}, \epsilon$
2: **Output:** Relevance weight vector $\mathbf{w}^*$
3: Initialize weight vector $\mathbf{w}^* \leftarrow \mathbf{0}_M$
4: **for** each mini-batch $\mathcal{B} \subset \mathcal{D}_{\mathrm{src}}$ **do**
5:     Construct the cost matrix $C$ between $\mathcal{B}$ and $\mathcal{D}_{\mathrm{tar}}$
6:     Define UOT geometry with entropic regularization $\epsilon$
7:     $\mathbf{w}_{\mathcal{B}} \leftarrow \mathrm{Sinkhorn}(C, \epsilon, \lambda_{\mathrm{src}}, \lambda_{\mathrm{tar}})$
8:     Update global weights: $\mathbf{w}^*[\mathcal{B}] \leftarrow \mathbf{w}_{\mathcal{B}}$
9: **end for**
10: **return** $\mathbf{w}^*$

---

**Algorithm 2** ROOT for Cross-Domain Offline RL

---

1: **Input:** Source dataset $\mathcal{D}_{\mathrm{src}}$, target dataset $\mathcal{D}_{\mathrm{tar}}$
2: **Hyperparameters:** $\lambda_{\mathrm{src}}, \lambda_{\mathrm{tar}}, \epsilon, \lambda_\beta$
3: **Output:** Target policy $\pi_\phi$
4: *// Phase 1: Cross-domain alignment*
5: Standardize transition features in $\mathcal{D}_{\mathrm{src}}$ and $\mathcal{D}_{\mathrm{tar}}$
6: Compute relevance weights $\mathbf{w}^*$ using Algorithm 1
7: Normalize weights: $\tilde{w}_i = \frac{w_i^*}{\frac{1}{M}\sum_{j=1}^M w_j^*}$
8: *// Phase 2: Weighted policy extraction*
9: Pretrain CVAE behavior model $\pi_\beta$ on $\mathcal{D}_{\mathrm{tar}}$
10: Initialize critic $Q_\theta$, value network $V_\psi$, and actor $\pi_\phi$
11: **for** each training iteration **do**
12:     Update critic $Q_\theta$ by minimizing the weighted TD loss (Eq. 12)
13:     Update value network $V_\psi$ via IQL expectile regression
14:     Update actor $\pi_\phi$ by minimizing the target-constrained actor loss (Eq. 13)
15: **end for**
16: **return** $\pi_\phi$

---

grounded, exhibit exponential outlier suppression (Proposition 3.2), and enable downstream policy learning to focus on geometrically relevant source transitions with quantifiable guarantees.

# 4. Algorithm

Having derived the theoretical formulation and guarantees of ROOT, we now present its algorithmic instantiation. The framework operates in two decoupled phases: (1) an alignment phase that solves the regularized UOT problem to compute relevance weights $\mathbf{w}^*$ once as preprocessing; and (2) a policy extraction phase that integrates the learned weights into an offline RL backbone. This modular design allows the data-dependent alignment to be precomputed once, enabling stable and efficient policy learning that focuses on transferable source transitions.

## 4.1. Scalable UOT Solver

To solve Eq. (9) efficiently on large-scale datasets, we employ an entropically regularized formulation implemented via the Sinkhorn algorithm (Cuturi, 2013). Specifically, we introduce an entropic regularization term $\epsilon H(\Gamma)$ to the objective. This enables the use of the Sinkhorn-Knopp algorithm, which solves the problem via efficient matrix-vector multiplications on GPUs.

In our implementation, we utilize the JAX framework and OTT library (Cuturi et al., 2022) to leverage hardware acceleration. Specifically, we define the transport geometry using Euclidean costs $C_{ij} = \|x_i - y_j\|_2^2$ on standardized transition features. The algorithm iteratively updates dual potentials to enforce the relaxed marginal constraints (controlled by $\lambda_{\mathrm{src}}, \lambda_{\mathrm{tar}}$). Upon convergence, the optimal relevance weights are obtained directly from the source marginal of the transport plan:

$$w^* = M\,\Gamma^*\mathbf{1} = M \cdot [\mathbf{u} \odot (K\mathbf{v})]. \qquad (11)$$

**Block-Wise Computation for Scalability.** Although JAX provides significant speedups, constructing the full $M \times N$ cost matrix is memory-prohibitive for large offline datasets (where $M \approx 10^6$). We therefore employ a block-wise strategy, as shown in Algorithm 1. We partition the source dataset into mini-batches $\{\mathcal{B}_k\}$ (e.g., $|\mathcal{B}_k| = 10^4$). For each batch, we construct the kernel matrix against the full target dataset, solve the UOT sub-problem, and extract the corresponding weights. This approximation is motivated by the locality of outlier rejection in UOT: the decision to discard an outlier depends primarily on its individual transport cost relative to $\lambda_{\mathrm{src}}$, rather than global competition. This design ensures linear scalability with respect to the source dataset size.

## 4.2. Weighted Offline Policy Extraction

With the relevance weights $w^*$ computed, we integrate them into an offline RL backbone. We adopt Implicit Q-Learning (IQL) (Kostrikov et al., 2022) as our backbone due to its stability. ROOT extends IQL by using the learned relevance weights to reweight source transitions during policy optimization.

**Weighted Critic Update.** We modify the IQL critic objective to incorporate source data. Specifically, the critic $Q_\theta$ is trained using target transitions together with reweighted source transitions, where the learned relevance weight $\tilde{w}$ modulates the contribution of each source transition:

$$L_Q(\theta) = \mathbb{E}_{\mathcal{D}_{\mathrm{tar}}}[L_{\mathrm{IQL}}] + \mathbb{E}_{\mathcal{D}_{\mathrm{src}}}[\tilde{w}(s,a,s') \cdot L_{\mathrm{IQL}}]. \quad (12)$$

where $L_{\mathrm{IQL}} = (r + \gamma V(s') - Q(s,a))^2$ is the standard Bellman regression loss used in IQL and the value function $V_\psi$ is updated by the standard IQL expectile regression loss.

Here, $\tilde{w}_i = \frac{w_i^*}{\frac{1}{M}\sum_{j=1}^{M} w_j^*}$ is the weight obtained after global mean normalization, which stabilizes gradient magnitudes during policy training. The value function $V_\psi$ and actor $\pi_\phi$ are updated following the standard IQL procedure, except that the actor is additionally constrained by the target behavior model as described below.

**Dataset-Constrained Policy Learning.** While the critic weights filter out geometrically incompatible transitions, the actor update can still be influenced by source-domain actions that are unsupported or kinematically infeasible in the target domain. To encourage target-compatible actions, we pretrain a conditional VAE (CVAE) to model the target behavior distribution $\pi_\beta(\cdot|s)$ using only target samples. We then regularize the actor $\pi_\phi$ toward the target behavior support:

$$\mathcal{L}_\pi(\phi) = \mathcal{L}_{\text{IQL}}(\phi) - \lambda_\beta \cdot \mathbb{E}_{s\sim\mathcal{D}_{\text{tar}}\cup\mathcal{D}_{\text{src}}}\left[\log \pi_\beta(\pi_\phi(s)|s)\right], \tag{13}$$

where $\mathcal{L}_{\text{IQL}}$ is the standard IQL advantage-weighted regression loss, and $\lambda_\beta > 0$ controls the strength of the target-behavior constraint. This regularization ensures that while the policy learns from the diverse source dynamics (via the critic), its output actions remain within the supported manifold of the target domain. Algorithm 2 summarizes the complete ROOT procedure.

# 5. Experiments

In this section, we evaluate the effectiveness of ROOT on cross-domain offline RL benchmarks with diverse dynamics shifts. Our experiments are designed to answer four key questions: (i) Does ROOT outperform state-of-the-art baselines across datasets of varying quality, especially in data-scarce target regimes? (ii) As the degree of domain shift increases, does ROOT remain consistently superior to other methods? (iii) Is explicit data filtering necessary during downstream policy learning? (iv) How sensitive is ROOT to the key hyperparameters, $\lambda_{\text{src}}$ and $\lambda_{\text{tar}}$?

## 5.1. Main Results

**Experimental and Dataset Setup.** To answer the first question, we evaluate ROOT on four continuous control environments (HalfCheetah, Hopper, Walker2d, Ant) under three distinct dynamics shifts: **gravity, kinematic, and morphology**. Detailed configurations are provided in Appendix D. For policy learning, we use standard D4RL (Fu et al., 2020) datasets with the '-v2' suffix as the source-domain data, covering three quality levels: medium, medium-replay, and medium-expert. For the target domain, we adopt the publicly released cross-domain offline datasets from OTDF (Lyu et al., 2025) to ensure fair comparison and reproducibility. Each target dataset contains only 5 trajectories ($\approx 5000$ transitions) per task and covers three quality levels: medium,

medium-expert, and expert.

**Baselines and Metric.** We compare ROOT against state-of-the-art cross-domain offline RL methods, including IQL (Kostrikov et al., 2022), which trains a policy on mixed source- and target-domain data; DARA (Liu et al., 2022), which estimates the dynamics gap using domain classifiers and penalizes source-domain rewards accordingly; BOSA (Liu et al., 2024), which explicitly learns a parameterized target-domain behavior policy to filter mixed datasets; SRPO (Xue et al., 2023), which leverages the stationary state distribution as a regularizer for reward modification; IGDF (Wen et al., 2024), which filters source-domain data using learned domain mutual information; OTDF (Lyu et al., 2025), which filters source-domain data based on deviation scores obtained from the optimal transport plan; and DmC (Le Pham Van et al., 2025), which both filters source-domain data and generates new data using a diffusion model. We evaluate performance using the standard D4RL normalized score, computed as $S_{\text{norm}} = 100 \times \frac{R-R_{\text{random}}}{R_{\text{expert}}-R_{\text{random}}}$, where $R$ is the average return of the learned policy, and $R_{\text{random}}$ and $R_{\text{expert}}$ are the reference returns of a random policy and an expert policy in the target domain, respectively. Reference values for each environment and shift type are provided in Appendix D.3. We report the performance of baseline methods using the best results reported from (Lyu et al., 2025) and (Le Pham Van et al., 2025).

**Performance Analysis.** We summarize the performance of ROOT and baselines under kinematic shifts in Table 1. Due to space limits, the full results under gravity and morphology shifts are deferred to Appendix F.1.

ROOT demonstrates superior performance across a broad range of tasks, achieving the highest normalized scores in 19 out of 36 scenarios. In terms of aggregate performance, ROOT achieves a total score of 2073.5, surpassing the strongest baseline, DmC, by a relative margin of 9.0%, and outperforming the balanced-OT baseline, OTDF (1547.6), by 34.0%. These results suggest that ROOT is an effective and robust solver for diverse dynamics mismatches without requiring manual threshold tuning. The most informative comparison arises in scenarios where the target dataset is high quality. As shown by the results on the Hopper-M $\rightarrow$ Expert task, OTDF suffers a sharp performance degradation, whereas ROOT achieves near-expert performance. This validates our core hypothesis: ROOT's transport-or-discard mechanism identifies poorly aligned medium-quality source transitions and suppresses them, while preserving the expert-aligned support.

## 5.2. Performance Under Varying Shift Levels

**Experimental Setting.** To answer the second question, we further evaluate ROOT on the ODRL benchmark under varying gravity and friction shift levels. Following

*Table 1.* **Performance comparison on Kinematic Shift tasks.** We report normalized scores and standard deviations over 5 seeds. Kinematic shifts modify physical capabilities such as joint or body-part properties. Source abbreviations use the form Env-Type: Ant/Half/Hopp/Walk denote Ant, HalfCheetah, Hopper, and Walker2d, respectively; m, me, and mr denote medium, medium-expert, and medium-replay source datasets. The Target column denotes the target-domain dataset quality. The best score in each setting is highlighted in bold.

| Source | Target | IQL | DARA | BOSA | SRPO | IGDF | OTDF | DmC | **ROOT** |
|---|---|---|---|---|---|---|---|---|---|
| Ant-m | medium | 50.0±5.6 | 42.3±7.6 | 20.9±2.6 | 50.5±6.7 | 54.5±1.3 | 55.4±0.0 | **62.1±0.6** | 61.9±0.3 |
| Ant-m | med-expert | 57.8±7.2 | 54.1±3.8 | 31.7±7.0 | 54.9±1.3 | 54.5±4.6 | 60.7±3.6 | **68.9±1.0** | 67.4±1.1 |
| Ant-m | expert | 59.6±18.5 | 54.2±11.3 | 45.4±8.6 | 45.5±9.3 | 49.4±14.6 | 90.4±4.8 | 92.1±3.5 | **97.2±0.8** |
| Ant-me | medium | 49.5±4.1 | 44.7±4.3 | 19.0±8.0 | 41.3±8.1 | 41.8±8.8 | 50.2±4.3 | **60.6±1.3** | 60.5±1.5 |
| Ant-me | med-expert | 37.2±2.0 | 33.3±7.0 | 6.4±2.5 | 32.8±8.0 | 41.5±4.9 | 48.8±2.7 | 60.4±3.7 | **65.4±2.8** |
| Ant-me | expert | 18.7±8.1 | 17.8±23.6 | 14.5±9.0 | 35.2±15.5 | 14.4±22.9 | 78.4±12.2 | 76.0±4.1 | **90.2±7.0** |
| Ant-mr | medium | 43.7±4.6 | 42.0±5.4 | 19.0±1.8 | 45.3±5.1 | 41.4±5.0 | 52.8±4.4 | **61.9±0.5** | 60.8±1.2 |
| Ant-mr | med-expert | 36.5±5.9 | 36.0±6.7 | 19.1±1.6 | 36.2±6.6 | 37.2±4.7 | 54.2±5.2 | 58.8±3.6 | **63.8±3.8** |
| Ant-mr | expert | 24.4±4.8 | 22.1±0.4 | 19.5±0.8 | 27.1±3.7 | 24.3±2.8 | **74.7±10.5** | 43.8±2.6 | 55.2±7.7 |
| Half-m | medium | 12.3±1.2 | 10.6±1.2 | 8.3±1.2 | 16.8±4.2 | 23.6±5.7 | 40.2±0.0 | 38.5±1.4 | **40.9±0.5** |
| Half-m | med-expert | 10.8±1.9 | 12.9±2.8 | 8.7±1.3 | 10.3±2.7 | 9.8±2.4 | 10.1±4.0 | 19.1±1.0 | **29.2±4.4** |
| Half-m | expert | 12.6±1.7 | 12.1±1.0 | 10.8±1.7 | 12.2±0.9 | 12.8±0.7 | 8.7±2.0 | **13.1±0.8** | 12.4±1.0 |
| Half-me | medium | 21.8±6.5 | 25.9±7.4 | 30.0±4.3 | 17.2±3.3 | 21.9±6.5 | 30.7±9.6 | 38.4±1.4 | **40.4±0.7** |
| Half-me | med-expert | 7.6±1.4 | 9.5±4.2 | 6.8±2.9 | 9.6±2.4 | 8.9±3.3 | 10.9±4.2 | 24.1±4.6 | **29.0±4.5** |
| Half-me | expert | 9.1±2.4 | 10.4±1.3 | 4.9±3.2 | 11.2±1.0 | 10.7±1.4 | 3.2±0.6 | **13.4±2.0** | 10.7±1.8 |
| Half-mr | medium | 10.0±5.4 | 11.5±4.9 | 7.5±3.1 | 10.2±3.7 | 11.6±4.6 | **37.8±2.1** | 19.5±1.8 | 30.4±4.9 |
| Half-mr | med-expert | 6.5±3.1 | 9.2±4.7 | 6.6±1.7 | 9.5±1.8 | 8.6±2.3 | 9.7±2.0 | **11.4±2.1** | 10.1±1.7 |
| Half-mr | expert | 13.6±1.4 | 14.8±2.0 | 10.4±4.9 | 14.8±2.2 | 13.9±2.2 | 7.2±1.4 | 15.6±2.9 | **21.4±3.6** |
| Hopp-m | medium | 58.7±8.4 | 43.9±15.2 | 12.3±6.6 | 65.4±1.5 | 65.3±1.4 | 65.6±1.9 | **69.8±2.3** | 68.2±0.5 |
| Hopp-m | med-expert | 68.5±12.4 | 55.4±16.9 | 15.6±10.8 | 43.9±30.8 | 51.1±18.5 | 55.4±25.1 | 78.2±5.1 | **90.3±13.7** |
| Hopp-m | expert | 79.9±35.5 | 83.7±19.6 | 14.8±5.5 | 53.1±39.8 | 87.4±25.4 | 35.0±19.4 | 59.8±21.8 | **99.4±2.0** |
| Hopp-me | medium | 66.0±0.5 | 61.1±4.0 | 35.0±20.1 | 64.6±2.6 | 65.2±1.5 | 65.3±2.4 | **69.6±1.3** | 68.2±0.9 |
| Hopp-me | med-expert | 45.1±15.7 | 61.9±16.9 | 13.9±4.9 | 54.7±17.0 | 62.9±15.6 | 38.6±15.9 | 75.5±9.6 | **86.8±9.3** |
| Hopp-me | expert | 44.9±19.8 | 84.2±21.1 | 12.0±4.3 | 57.6±40.6 | 52.8±19.7 | 29.9±11.3 | 64.5±24.2 | **92.3±19.7** |
| Hopp-mr | medium | 36.0±0.1 | 39.4±7.2 | 3.2±2.6 | 36.1±0.2 | 35.9±2.4 | 35.5±12.2 | **64.8±2.4** | 54.9±7.7 |
| Hopp-mr | med-expert | 36.1±0.1 | 34.1±3.6 | 4.4±2.8 | 36.0±0.1 | 36.1±0.1 | 47.5±14.6 | **69.7±7.5** | 60.4±9.8 |
| Hopp-mr | expert | 36.0±0.1 | 36.1±0.2 | 3.7±2.5 | 36.1±0.1 | 36.1±0.3 | 49.9±30.5 | **69.9±18.0** | 56.8±15.3 |
| Walk-m | medium | 34.3±9.8 | 35.2±22.5 | 14.3±11.2 | 39.0±6.7 | 41.9±11.2 | 49.6±18.0 | 63.2±4.2 | **64.3±6.3** |
| Walk-m | med-expert | 30.2±12.5 | 51.9±11.5 | 13.6±7.7 | 38.6±6.5 | 42.3±19.3 | 43.5±16.4 | 53.5±7.0 | **70.8±11.6** |
| Walk-m | expert | 56.4±18.2 | 40.7±14.4 | 15.3±2.5 | 57.3±12.2 | 60.4±17.5 | 46.7±13.6 | 70.5±12.0 | **90.5±8.9** |
| Walk-me | medium | 41.8±8.8 | 38.1±14.4 | 21.4±8.3 | 36.9±4.3 | 41.2±13.0 | 44.6±6.0 | 59.4±6.8 | **60.9±5.2** |
| Walk-me | med-expert | 22.2±8.7 | 23.6±8.1 | 15.9±4.1 | 23.2±7.9 | 28.1±4.0 | 16.5±7.2 | 53.2±7.3 | **53.5±5.0** |
| Walk-me | expert | 26.3±10.4 | 36.0±9.2 | 18.5±3.6 | 40.9±9.6 | 46.2±19.4 | 42.4±9.1 | 69.2±7.0 | **75.8±6.7** |
| Walk-mr | medium | 11.5±7.1 | 12.5±4.3 | 1.9±2.1 | 14.3±3.1 | 22.2±5.2 | 49.7±9.7 | **52.9±8.4** | 52.6±1.9 |
| Walk-mr | med-expert | 9.7±3.8 | 11.2±5.0 | 4.6±3.0 | 4.2±5.1 | 7.6±4.9 | **55.9±17.1** | 36.4±5.4 | 35.5±7.3 |
| Walk-mr | expert | 7.7±4.8 | 7.4±2.4 | 3.6±1.5 | 13.2±8.5 | 7.5±2.1 | **51.9±7.9** | 44.4±8.5 | 45.4±15.9 |
| **Total Score** | | 1193.0 | 1219.8 | 513.5 | 1195.7 | 1271.0 | 1547.6 | 1902.2 | **2073.5** |

the experimental protocol used in MOBODY (Guo et al., 2026), we also evaluate ROOT on four MuJoCo environments. The source domain is kept unchanged, while the target domain is modified by scaling the corresponding physical parameter. We report results on the common shift levels $\{0.1, 0.5, 2.0, 5.0\}$ for which all compared methods are available. We report the performance of baseline methods using the best results reported from (Guo et al., 2026).

**Performance Analysis.** As summarized in Table 2, ROOT achieves the best aggregate performance across the 32 gravity/friction shift settings, outperforming all compared cross-domain and off-dynamics offline RL baselines. In terms of total normalized score, ROOT obtains 1477.97, outperforming the strongest baseline MOBODY (1427.26) and REAG (986.13). ROOT achieves the highest score in 16 out of 32 settings and ranks within the top two in 26 out of 32 settings, demonstrating strong and stable performance across different environments, shift types, and shift magnitudes.

These results highlight ROOT's ability to adaptively select source transitions under different degrees of source-target overlap. At moderate shift levels such as 0.5 and 2.0, ROOT achieves strong aggregate performance. In particular, at shift level 2.0, ROOT obtains the best score in 7 out of 8 settings, suggesting that the transport-or-discard mechanism is especially effective when a substantial portion of source transitions becomes incompatible but useful transferable support still remains. At the more extreme shift levels 0.1 and 5.0, ROOT remains highly competitive: it achieves the best score in 8 out of 16 settings and ranks within the top two in 14 out of 16 settings.

### 5.3. Parameter Studies

To answer Questions (iii) and (iv), we study whether additional hard filtering is necessary during downstream policy learning and evaluate the sensitivity of ROOT to the key relaxation parameters $\lambda_{\text{src}}$ and $\lambda_{\text{tar}}$. For the filtering study, after obtaining the normalized source weight $\tilde{w}_i$, we

*Table 2.* Performance comparison with ODRL baselines under gravity and friction shifts. We report normalized target-domain scores and standard deviations. The best score in each setting is highlighted in bold.

| Env | Shift | BOSA | DARA | REAG | SRPO | MOBODY | **ROOT** |
|---|---|---|---|---|---|---|---|
| Half-Gra | 0.1 | 9.31±1.94 | 12.90±1.01 | 16.14±0.66 | **32.94±1.65** | 14.18±1.06 | 16.33±1.14 |
| Half-Gra | 0.5 | 43.96±5.68 | 46.11±1.93 | 40.50±1.58 | 41.99±1.63 | **47.18±1.23** | 39.53±11.36 |
| Half-Gra | 2.0 | 27.86±0.94 | 31.85±1.31 | 33.28±3.16 | 32.24±1.97 | 41.60±7.35 | **43.57±12.73** |
| Half-Gra | 5.0 | 17.95±11.97 | 27.67±17.01 | 71.31±2.80 | -2.33±0.69 | **83.05±1.21** | 74.17±0.35 |
| Half-Fri | 0.1 | 12.53±3.61 | 23.69±16.46 | 9.74±0.46 | 17.36±0.73 | **57.53±2.49** | 55.35±5.44 |
| Half-Fri | 0.5 | 68.93±0.35 | 64.89±3.04 | 66.50±0.99 | **109.18±2.15** | 69.54±0.48 | 67.01±1.13 |
| Half-Fri | 2.0 | 46.53±0.37 | 46.25±2.36 | 37.74±2.35 | **75.19±1.54** | 50.02±3.26 | 47.22±0.76 |
| Half-Fri | 5.0 | 44.07±9.07 | 40.06±7.87 | 25.74±3.24 | 5.10±1.96 | **59.20±4.91** | 56.89±11.61 |
| Ant-Gra | 0.1 | 25.58±2.21 | 11.03±1.24 | 15.75±1.17 | 13.78±1.81 | **37.09±2.12** | 32.83±2.91 |
| Ant-Gra | 0.5 | 19.03±4.41 | 9.04±1.35 | 13.25±0.86 | 7.02±2.73 | **37.44±2.79** | 29.79±6.25 |
| Ant-Gra | 2.0 | 41.77±1.52 | 36.64±0.82 | 43.25±1.72 | 4.17±2.03 | 45.83±1.71 | **58.23±12.14** |
| Ant-Gra | 5.0 | 31.94±0.69 | 31.01±0.39 | 49.36±2.61 | 8.45±1.24 | 65.45±3.23 | **70.74±0.73** |
| Ant-Fri | 0.1 | **58.95±0.71** | 55.12±0.24 | 54.13±0.56 | 2.55±3.45 | 58.79±0.11 | 55.60±0.14 |
| Ant-Fri | 0.5 | 59.72±3.57 | 58.92±0.80 | 57.46±0.65 | 6.57±1.76 | **62.41±4.10** | 59.92±0.01 |
| Ant-Fri | 2.0 | 20.18±3.79 | 17.54±2.47 | 21.28±0.72 | 10.81±2.09 | 47.41±4.40 | **55.96±10.49** |
| Ant-Fri | 5.0 | 9.07±0.88 | 7.80±0.12 | 9.53±0.65 | 11.72±1.86 | 31.17±5.57 | **35.87±11.60** |
| Wal-Gra | 0.1 | 18.75±12.02 | 20.12±5.74 | 26.56±2.62 | 13.67±3.19 | 65.85±5.08 | **75.48±2.95** |
| Wal-Gra | 0.5 | 40.09±20.37 | 29.72±16.02 | 55.20±2.18 | **56.28±2.34** | 43.57±2.32 | 51.79±5.42 |
| Wal-Gra | 2.0 | 8.91±2.28 | 32.20±1.05 | 13.50±2.38 | 8.52±0.82 | 44.32±4.58 | **61.52±3.97** |
| Wal-Gra | 5.0 | 5.25±0.50 | 5.44±0.08 | 4.61±1.13 | 5.12±0.46 | **46.05±20.73** | 5.23±0.18 |
| Wal-Fri | 0.1 | 7.88±1.88 | 5.65±0.06 | 10.58±0.71 | 9.02±0.81 | 28.23±9.13 | **48.11±6.44** |
| Wal-Fri | 0.5 | 63.94±20.40 | 68.81±1.12 | **78.58±1.08** | -0.23±0.45 | 76.96±1.99 | 77.20±10.08 |
| Wal-Fri | 2.0 | 39.06±17.36 | 72.91±0.37 | 42.18±3.85 | 15.51±0.73 | 73.74±0.49 | **73.85±0.47** |
| Wal-Fri | 5.0 | 10.07±4.91 | 5.36±0.28 | 8.36±1.91 | 4.94±0.66 | 27.38±3.87 | **33.62±12.54** |
| Hop-Gra | 0.1 | 27.82±13.41 | 23.40±11.62 | 31.11±1.80 | 17.62±1.66 | 36.25±1.50 | **48.21±9.87** |
| Hop-Gra | 0.5 | 28.54±12.77 | 12.86±0.18 | 36.37±2.06 | **67.06±3.60** | 33.57±6.71 | 49.03±8.18 |
| Hop-Gra | 2.0 | 11.84±2.37 | 14.65±2.47 | 16.44±1.60 | 12.09±0.71 | 23.79±2.09 | **27.21±2.53** |
| Hop-Gra | 5.0 | 7.36±0.13 | 7.90±1.27 | 8.11±0.97 | 7.48±0.51 | 8.06±0.03 | **8.84±0.01** |
| Hop-Fri | 0.1 | 25.55±2.69 | 26.13±4.24 | 33.08±2.53 | 18.21±0.85 | **51.19±2.56** | 51.15±12.47 |
| Hop-Fri | 0.5 | 25.22±4.48 | 26.94±2.86 | 38.10±3.32 | 18.41±1.31 | 41.34±0.49 | **48.39±10.78** |
| Hop-Fri | 2.0 | 10.32±0.06 | 10.15±0.03 | 10.20±0.30 | 9.71±0.37 | 11.00±0.14 | **11.05±0.04** |
| Hop-Fri | 5.0 | 7.90±0.06 | 7.86±0.05 | 8.20±0.36 | 7.76±0.26 | 8.07±0.04 | **8.26±0.07** |
| **Total Score** | | 875.88 | 890.62 | 986.13 | 647.91 | 1427.26 | **1477.97** |

introduce an additional threshold-based filtering step that discards source transitions with $\tilde{w}_i < \tau_{\text{th}}$. We vary the threshold as $\tau_{\text{th}} \in \{0.0, 0.5, 1.0, 1.5, 2.0\}$, where $\tau_{\text{th}} = 0$ corresponds to no additional hard filtering. For the relaxation parameters, we sweep five values for each of $\lambda_{\text{src}}$ and $\lambda_{\text{tar}}$ within $[0.01, 2.0]$. We conduct these studies on four representative environments, including halfcheetah-gravity, halfcheetah-morph, hopper-kinematic, and walker2d-morph, using medium-quality datasets in both source and target domains.

**Robustness to the Filtering Threshold.** As shown in Figure 2, ROOT exhibits a stable performance plateau across a wide range of $\tau_{\text{th}}$, and introducing additional hard filtering yields no consistent improvement. This supports the effectiveness of the proposed transport-or-discard mechanism: geometrically incompatible source transitions are already assigned near-zero relevance weights by ROOT, so further thresholding provides limited benefit for downstream policy learning.

**Sensitivity to Marginal Relaxations.** Figure 3 shows that ROOT has a broad effective range for the UOT relaxation parameters. For $\lambda_{\text{src}}$, ROOT remains stable over a wide range of values. When $\lambda_{\text{src}}$ is too small, the source-side penalty becomes overly weak and ROOT may discard useful transferable samples; when it is too large, the objective

approaches balanced OT and may force incompatible source mass to be transported. Moderate values therefore provide a better trade-off between preserving transferable source data and suppressing incompatible transitions. ROOT is even less sensitive to $\lambda_{\text{tar}}$: the learning curves for $\lambda_{\text{tar}} \in [0.05, 2.0]$ are tightly clustered and often nearly indistinguishable. This suggests that the target-side relaxation has a broad stable range, and once sufficient relaxation is allowed to account for limited target samples, the exact value of $\lambda_{\text{tar}}$ has only a minor influence on policy quality.

## 6. Conclusion

In this paper, we proposed ROOT, a robust unbalanced transport framework for cross-domain offline RL that computes absolute availability scores for source transitions under dynamics mismatch. By relaxing the strict mass conservation of balanced optimal transport, ROOT yields a principled transport-or-discard mechanism that selectively aligns source data based on geometric compatibility, mitigating the forced matching effect that can amplify negative transfer under partial overlap. On the theoretical side, we established that the relevance weights induced by ROOT exhibit exponential suppression with respect to transport cost, providing a formal guarantee for outlier down-weighting. Empirically, with a scalable block-wise Sinkhorn implementation, ROOT

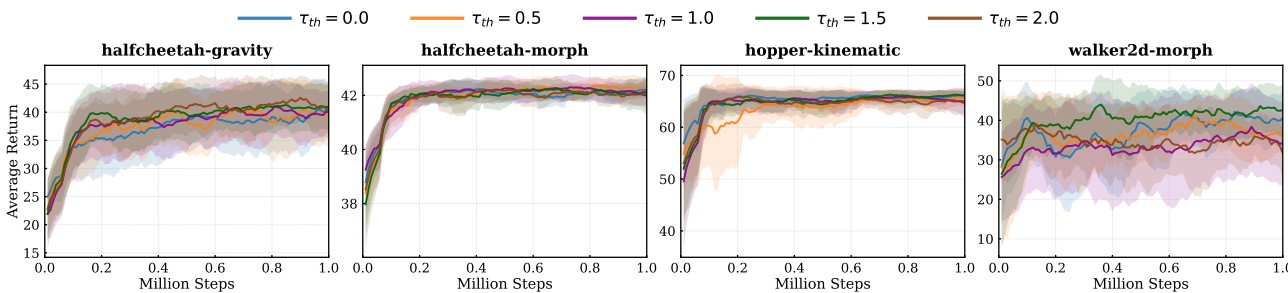

*Figure 2.* Parameter Study on filter threshold $\tau_{\text{th}}$. The solid lines depict the average returns over 5 seeds and the shaded area denotes the standard deviation.

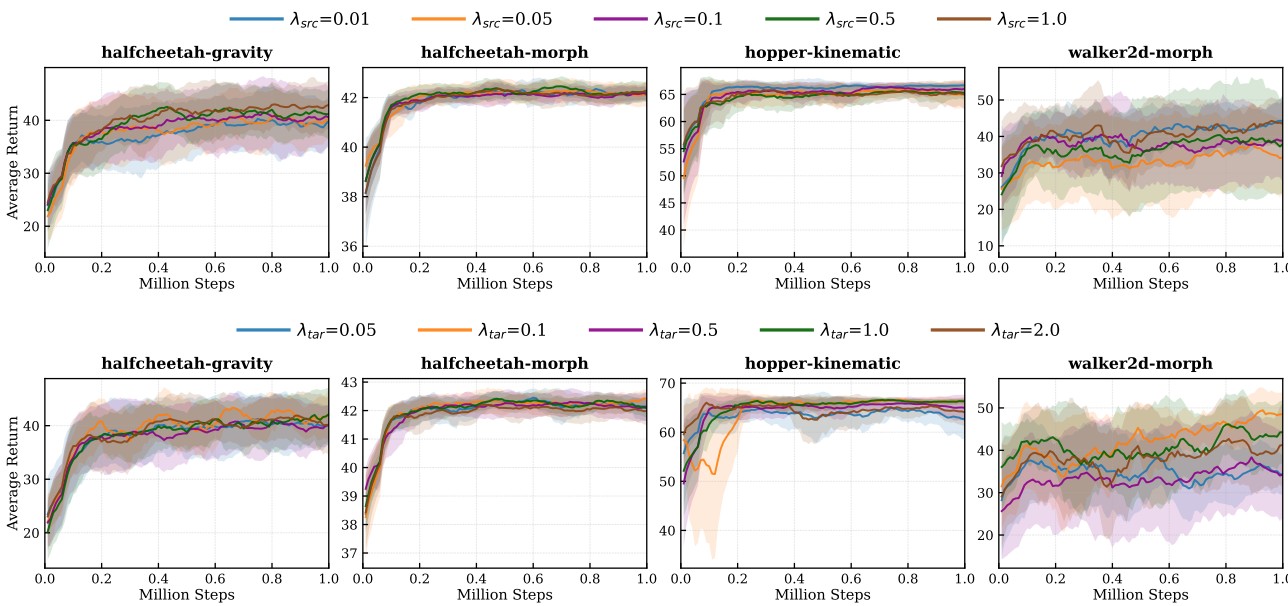

*Figure 3.* **Parameter Sensitivity Analysis on $\lambda_{\text{src}}$ and $\lambda_{\text{tar}}$.** The solid lines depict the average returns over 5 seeds and the shaded area denotes the standard deviation

consistently improves downstream offline RL and outperforms strong baselines across diverse D4RL dynamics shifts, showing strong stability in data-scarce and large-mismatch regimes.

**Limitations**. ROOT depends on the choice of geometric cost (and representation) for measuring cross-domain compatibility, and transport computation may still be non-trivial for very large-scale or high-dimensional settings. Future work will investigate learned or adaptive costs and more efficient transport solvers, as well as tighter integration with policy learning for joint alignment and control.

# Acknowledgements

This work was supported in part by the construction fund for the Changzhi City "Open Competition" Project (Key Technologies and Demonstration for the Construction of Green Airport and Intelligent Management & Control Platform), Changzhi City Key Research and Development Project (Smart Sports Park Management System and Supporting Design and Development), the Major Science and Technology Project of Shanxi Province (Grant No. 202101010101018), the Young Scientists Fund of the National Natural Science Foundation of China (No. 62506154), the Fundamental Research Program of Shanxi Province (No. 20210302123025), the Key Basic Research Program of Shanxi Province (No. 202102010101011), and the Joint Funds of the National Natural Science Foundation of China (No. U21A20524).

# Impact Statement

This paper presents work whose goal is to advance the field of Machine Learning. There are many potential societal consequences of our work, none which we feel must be

specifically highlighted here.

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

# A. Related Work

**Offline Reinforcement Learning** Offline RL (or batch RL) (Levine et al., 2020) aims to learn optimal policies solely from static, previously collected datasets without any further interaction with the environment. The fundamental challenge in this setting is the distributional shift between the learned policy and the behavior policy generating the dataset, which often leads to erroneous value overestimation for out-of-distribution (OOD) state-actions. Existing methods can be generally categorized into model-free and model-based approaches (Prudencio et al., 2023). Model-free algorithms typically address the OOD challenge through three main mechanisms, including policy constraint methods (Fujimoto et al., 2019; Fujimoto & Gu, 2021; Kostrikov et al., 2022; Wu et al., 2022; Ran et al., 2023; Tarasov et al., 2023) which explicitly or implicitly constrain the learned policy to stay close to the behavior policy distribution; value penalization methods (Kumar et al., 2020; Lyu et al., 2022; Nikulin et al., 2023; Yeom et al., 2024), which learn a conservative value function by penalizing Q-values of OOD actions to lower-bound the true value; and uncertainty quantification methods (Wu et al., 2021; An et al., 2021), which reweight updates based on the epistemic uncertainty of the value estimates. Alternatively, model-based offline RL methods (Yu et al., 2020; Kidambi et al., 2020; Rigter et al., 2022; Guo et al., 2022; Qiao et al., 2025; 2026b) learn a dynamics model from the dataset and use it to generate synthetic rollouts for policy optimization, often incorporating penalization on the model's uncertainty to prevent exploiting model errors. However, the success of these standard offline RL methods relies heavily on the assumption that the training and testing environments share identical dynamics and that the static dataset contains a sufficient coverage of transitions. Their performance degrades catastrophically when transferred to a target domain with dynamics mismatch, necessitating specialized cross-domain adaptation techniques.

**Dynamic Adaptation in RL.** Achieving policy generalization across varying dynamics remains a fundamental challenge in reinforcement learning(Niu et al., 2024; Guo et al., 2024; Van et al., 2025). Traditional approaches typically rely on system identification (Du et al., 2021; Xie et al., 2022; Fernando et al., 2013; Chebotar et al., 2019; Werbos, 1989) to capture the physical and visual properties of the real world, domain randomization (Tobin et al., 2017; Peng et al., 2018; Akkaya et al., 2019; Jiang et al., 2023) which introduces diversity by randomly altering simulation parameters, meta-RL (Finn et al., 2017; Clavera et al., 2019) for rapid policy fine-tuning and imitation learning (Chae et al., 2022; Kim et al., 2020; Liu et al., 2023) for expert-level policy extraction. However, these methods often necessitate expert demonstrations from the target domain or require extensive prior knowledge to guide parameter randomization.More recently, research has expanded to relax the strict offline constraint by assuming access to an auxiliary domain with discrepant dynamics. These works typically operate under hybrid settings, such as **offline-to-online adaptation** (Kong et al., 2025) where shifted offline data facilitates online target learning, **sim-to-real transfer** (Chen et al., 2025; Niu et al., 2022; 2025) which combines an imperfect online simulator with sparse real-world data, or **adaptation with limited interactions** (Lyu et al., 2024a; Xu et al., 2023; Eysenbach et al., 2021; Le Pham Van et al., 2024) where abundant source data is paired with a few online trajectories from the target domain.

Unlike these approaches that rely on partial online access or interactive feedback, we focus on the *offline policy adaptation* setting where both domains are strictly offline (Qiao et al., 2026c;d). Existing solutions to this problem primarily fall into two paradigms: **parametric approaches** and **non-parametric approaches**.The dominant parametric paradigm involves training high-capacity models, such as deep neural networks, to explicitly capture domain discrepancies. Prominent examples include training dynamics-aware discriminators for reward augmentation (Liu et al., 2022; Wang et al., 2026), estimating density ratios to enforce support constraints (Liu et al., 2024), learning contrastive representations for data filtering (Wen et al., 2024) and directly learning dynamics models from two domain datasets (Guo et al., 2026). However, a critical bottleneck of these methods is their sensitivity to the quantity of target domain data. When target transitions are extremely scarce, training such parameterized models inevitably leads to severe **overfitting** and **estimation bias**, resulting in unstable adaptation. To mitigate the reliance on model training and improve robustness in data-scarce regimes, recent research has shifted towards non-parametric methods that exploit intrinsic geometric or statistical measures. Representative works like DmC (Le Pham Van et al., 2025) utilize $k$-nearest neighbor estimation to measure domain divergence locally, while OTDF (Lyu et al., 2025) formulates global domain alignment as a standard Optimal Transport (OT) problem. Nevertheless, a fundamental limitation of these OT-based methods lies in their reliance on balanced transport, which strictly enforces mass conservation constraints (i.e., all source probability mass must be transported to the target). In cross-domain settings with significant dynamics shifts, this rigid constraint leads to the forced matching problem: source transitions with distinct dynamics are forcibly mapped to the target distribution to satisfy marginal constraints, thereby introducing incorrect bias and negative transfer.To address this, we propose ROOT, which fundamentally reformulates the alignment problem using Unbalanced Optimal Transport. By relaxing the marginal constraints, ROOT establishes a *transport-or-discard* mechanism that selectively aligns compatible transitions while automatically discarding outliers, ensuring robust adaptation even under severe domain mismatches.

# B. Detailed Derivation

## B.1. Detailed Derivation of Linear Programming Problem

In this section, we provide the detailed derivation showing that the soft subset selection objective (Eq.(3) in the main text) is equivalent to the joint linear programming problem (Eq.(4)).

**Lemma B.1** (Equivalence to Joint LP). *Let $\hat{\nu}_{\text{src}} = \sum_{i=1}^{M} \hat{\nu}_{\text{src},i}\delta_{x_i}$ and $\hat{\nu}_{\text{tar}} = \sum_{j=1}^{N} \hat{\nu}_{\text{tar},j}\delta_{y_j}$ be the empirical source and target distributions. The bi-level optimization problem:*

$$\min_{\mathbf{w}\in\Delta_w} W_c(\mathbf{w}\odot\hat{\nu}_{\text{src}}, \hat{\nu}_{\text{tar}}) \quad s.t. \quad \sum_{i=1}^{M} \hat{\nu}_{\text{src},i}w_i = 1, \ w_i \geq 0, \tag{14}$$

*is equivalent to the following single-level linear program over the transport plan $\Gamma \in \mathbb{R}_+^{M\times N}$:*

$$\min_{\Gamma\geq 0}\langle\Gamma, C\rangle \quad s.t. \quad \Gamma^\top\mathbf{1}_M = \hat{\nu}_{\text{tar}}. \tag{15}$$

*Proof.* First, expanding the definition of the Wasserstein distance $W_c$ in (14) using the Kantorovich formulation, we obtain the nested problem:

$$\min_{\mathbf{w}\geq 0}\left[\min_{\Gamma\geq 0}\langle\Gamma, C\rangle \quad \text{s.t.} \quad \Gamma\mathbf{1}_N = \mathbf{w}\odot\hat{\nu}_{\text{src}}, \quad \Gamma^\top\mathbf{1}_M = \hat{\nu}_{\text{tar}}\right] \quad \text{s.t.} \quad \sum_{i=1}^{M}\hat{\nu}_{\text{src},i}w_i = 1. \tag{16}$$

Since the inner minimization depends on $\mathbf{w}$ only through the constraint set, we can merge the optimizations into a joint minimization over $(\mathbf{w}, \Gamma)$:

$$\min_{\mathbf{w},\Gamma\geq 0}\langle\Gamma, C\rangle \quad \text{s.t.} \quad \begin{cases} (\Gamma\mathbf{1}_N)_i = w_i\hat{\nu}_{\text{src},i}, & \forall i \\ \Gamma^\top\mathbf{1}_M = \hat{\nu}_{\text{tar}} \\ \sum_i \hat{\nu}_{\text{src},i}w_i = 1 \end{cases} \tag{17}$$

**Variable Elimination.** We observe that the variable $w_i$ is fully determined by the source marginal of $\Gamma$. Assuming $\hat{\nu}_{\text{src},i} > 0$, we can uniquely express $w_i$ in terms of $\Gamma$:

$$w_i(\Gamma) = \frac{(\Gamma\mathbf{1}_N)_i}{\hat{\nu}_{\text{src},i}}. \tag{18}$$

Substituting (18) into the normalization constraint $\sum_i \hat{\nu}_{\text{src},i}w_i = 1$ yields:

$$\sum_{i=1}^{M}\hat{\nu}_{\text{src},i}\left(\frac{(\Gamma\mathbf{1}_N)_i}{\hat{\nu}_{\text{src},i}}\right) = \sum_{i=1}^{M}\sum_{j=1}^{N}\Gamma_{ij} = 1. \tag{19}$$

Thus, the normalization constraint on $\mathbf{w}$ is equivalent to a global mass constraint on $\Gamma$. The optimization problem simplifies to:

$$\min_{\Gamma\geq 0}\langle\Gamma, C\rangle \quad \text{s.t.} \quad \Gamma^\top\mathbf{1}_M = \hat{\nu}_{\text{tar}} \quad \text{and} \quad \sum_{i,j}\Gamma_{ij} = 1. \tag{20}$$

**Constraint Redundancy.** Note that since the target distribution is a probability measure, it satisfies $\sum_j \hat{\nu}_{\text{tar},j} = 1$. The constraint $\Gamma^\top\mathbf{1}_M = \hat{\nu}_{\text{tar}}$ implies:

$$\sum_{i,j}\Gamma_{ij} = \sum_{j=1}^{N}(\Gamma^\top\mathbf{1}_M)_j = \sum_{j=1}^{N}\hat{\nu}_{\text{tar},j} = 1. \tag{21}$$

Therefore, the global mass constraint is redundant. The problem reduces to optimizing $\Gamma$ subject only to the target marginal constraint, recovering Eq. (15).

Finally, given an optimal plan $\Gamma^*$, the optimal weights are recovered via Eq. (18), which matches Eq.(4) in the main text. $\square$

## B.2. Detailed Derivation of Target Relaxation

In this section, we provide the detailed mathematical derivation transforming the constrained joint minimization problem in (6) into the penalized objective function in (7).

**The Primal Constrained Problem.** Recall the formulation in (6), where we seek to jointly optimize the transport plan $\Gamma$ and the auxiliary target marginal $\nu$ within a robust uncertainty set $\mathcal{U}_\delta(\hat{\nu}_{\mathrm{tar}})$. The optimization problem is defined as:

$$
\min_{\Gamma \geq 0, \nu} \langle \Gamma, C \rangle \quad \text{s.t.} \quad
\begin{cases}
\Gamma^\top \mathbf{1} = \nu \\
D_{\mathrm{KL}}(\nu \| \hat{\nu}_{\mathrm{tar}}) \leq \delta \\
\sum_{i,j} \Gamma_{ij} = 1
\end{cases}
\tag{22}
$$

Here, the first constraint enforces consistency between the plan and the target marginal, the second defines the uncertainty set, and the third enforces the global mass conservation inherited from the soft subset selection problem.

**Lagrangian Relaxation.** To relax the inequality constraint $D_{\mathrm{KL}}(\nu \| \hat{\nu}_{\mathrm{tar}}) \leq \delta$, we introduce a Lagrange multiplier $\lambda_{\mathrm{tar}} \geq 0$. The Lagrangian function associated with this constraint is:

$$
\mathcal{L}(\Gamma, \nu, \lambda_{\mathrm{tar}}) = \langle \Gamma, C \rangle + \lambda_{\mathrm{tar}} \left( D_{\mathrm{KL}}(\nu \| \hat{\nu}_{\mathrm{tar}}) - \delta \right)
\tag{23}
$$

Under the assumption that Slater's condition holds (i.e., the feasible set is non-empty and has an interior), strong duality applies. This implies that for a specific uncertainty radius $\delta$, there exists a corresponding penalty coefficient $\lambda_{\mathrm{tar}}$ such that minimizing the constrained problem is equivalent to minimizing the penalized Lagrangian.

Dropping the constant term $-\lambda_{\mathrm{tar}}\delta$ (which does not depend on optimization variables $\Gamma$ or $\nu$), the problem becomes:

$$
\min_{\Gamma \geq 0, \nu} \langle \Gamma, C \rangle + \lambda_{\mathrm{tar}} D_{\mathrm{KL}}(\nu \| \hat{\nu}_{\mathrm{tar}}) \quad \text{s.t.} \quad \Gamma^\top \mathbf{1} = \nu, \quad \sum_{i,j} \Gamma_{ij} = 1
\tag{24}
$$

**Variable Substitution and Simplification.** We observe that the auxiliary variable $\nu$ is strictly determined by the transport plan $\Gamma$ via the linear equality constraint $\Gamma^\top \mathbf{1} = \nu$. This allows us to eliminate $\nu$ by directly substituting $\Gamma^\top \mathbf{1}$ into the KL-divergence term in the objective function.

Substituting $\nu = \Gamma^\top \mathbf{1}$ yields the final single-variable optimization problem:

$$
\min_{\Gamma \geq 0} \langle \Gamma, C \rangle + \lambda_{\mathrm{tar}} D_{\mathrm{KL}}(\Gamma^\top \mathbf{1} \| \hat{\nu}_{\mathrm{tar}}) \quad \text{s.t.} \quad \sum_{i,j} \Gamma_{ij} = 1
\tag{25}
$$

This strictly recovers the formulation in (7), completing the derivation.

## B.3. Derivation of Availability Scores from Optimal Plan

In this section, we derive the explicit formula for recovering the availability scores $w^*$ from the optimal transport plan $\Gamma^*$, as presented in Eq. (11).

**The Relationship between Plan and Weights.** Recall from the definition of the Soft Subset Selection problem (Definition 3.1) that we seek a weighting function $w$ such that the reweighted source distribution $\nu_w = w \odot \hat{\nu}_{\mathrm{src}}$ is geometrically close to the target.

In the Kantorovich formulation of Optimal Transport, the transport plan $\Gamma \in \mathbb{R}_+^{M \times N}$ represents the joint probability mass between the reweighted source distribution and the target distribution. A fundamental property of the valid coupling is that its first marginal (summing over target columns) must recover the source distribution being transported. Therefore, we have the identity:

$$
\Gamma \mathbf{1} = \nu_w = w \odot \hat{\nu}_{\mathrm{src}}
\tag{26}
$$

**Component-wise Derivation.** Considering the discrete empirical distributions, let the source dataset contain $M$ samples. For a specific source sample $x_i$ (where $i = 1, \ldots, M$), the relation in (26) can be written component-wise as:

$$
(\Gamma \mathbf{1})_i = w_i \cdot \hat{\nu}_{src,i}
\tag{27}
$$

Here, $(\Gamma\mathbf{1})_i = \sum_{j=1}^{N} \Gamma_{ij}$ represents the total mass transported away from the $i$-th source sample to any target sample.

Solving for $w_i$, we obtain the general recovery formula:

$$w_i = \frac{(\Gamma\mathbf{1})_i}{\hat{\nu}_{src,i}} = \frac{\sum_{j=1}^{N} \Gamma_{ij}}{\hat{\nu}_{src,i}} \tag{28}$$

**Uniform Prior Assumption.** In standard offline reinforcement learning settings, the empirical source distribution is assumed to be uniform over the dataset, i.e., $\hat{\nu}_{src} = \sum_{i=1}^{M} \frac{1}{M} \delta_{x_i}$. Thus, the probability mass for each sample is:

$$\hat{\nu}_{src,i} = \frac{1}{M}, \quad \forall i \in \{1, \dots, M\} \tag{29}$$

Substituting this into the general formula, we derive the final expression for the optimal availability scores $w^*$:

$$w_i^* = \frac{\sum_{j=1}^{N} \Gamma_{ij}^*}{1/M} = M \sum_{j=1}^{N} \Gamma_{ij}^* \tag{30}$$

This confirms that the availability score for a transition is strictly proportional to the total probability mass assigned to it by the optimal transport plan, scaled by the dataset size $M$.

## C. Detailed Proof of Proposition 3.2

In this section, we provide a detailed proof of the exponential decay property of the ROOT weights that are omitted from the main text due to space limit. The proof proceeds in three steps: (1) deriving the optimality conditions and dual form of the entropically regularized objective, (2) performing an asymptotic analysis as the entropy regularization vanishes, and (3) establishing bounds on the weights via the boundedness of the dual potentials.

To provide a rigorous proof for the exponential decay of relevance weights, we first establish the necessary regularity conditions regarding the geometry of the state space.

**Assumption C.1** (Regularity of Cost and Domain). Let the source domain $\mathcal{X}$ and target domain $\mathcal{Y}$ be compact subsets of $\mathbb{R}^d$. The ground cost function $C : \mathcal{X} \times \mathcal{Y} \to \mathbb{R}_{\geq 0}$ is assumed to be continuous and bounded. Consequently, there exists a finite constant $B_C < \infty$ such that $\|C\|_\infty \leq B_C$. Furthermore, we assume the empirical marginal distributions $\hat{\nu}_{src}$ and $\hat{\nu}_{tar}$ are strictly positive on their support.

We begin by deriving the optimality conditions for the entropically regularized objective and establishing the asymptotic behavior of the dual potentials.

**Lemma C.2** (Asymptotic Dual Convergence). *Consider the entropically regularized ROOT objective with regularization parameter $\epsilon > 0$. Let $(f^{(\epsilon)}, g^{(\epsilon)})$ denote the optimal dual potentials associated with the source and target marginals, respectively. In the limit as $\epsilon \to 0$, the source potential $f_i^{(\epsilon)}$ satisfies:*

$$\lim_{\epsilon \to 0} f_i^{(\epsilon)} = -\min_j \left( C_{ij} + g_j^{(0)} \right), \tag{31}$$

*where $g_j^{(0)}$ is the optimal target dual potential of the unregularized semi-discrete unbalanced optimal transport problem.*

*Proof.* The objective function $\mathcal{J}_\epsilon(\Gamma)$ is strictly convex. Setting the gradient with respect to the transport plan $\Gamma_{ij}$ to zero yields the first-order optimality condition:

$$C_{ij} + \lambda_{src} \log \left( \frac{(\Gamma\mathbf{1})_i}{\hat{\nu}_{src,i}} \right) + \lambda_{tar} \log \left( \frac{(\Gamma^\top\mathbf{1})_j}{\hat{\nu}_{tar,j}} \right) + \epsilon \log \Gamma_{ij} = 0. \tag{32}$$

We implicitly define the $\epsilon$-dependent dual potentials $f_i^{(\epsilon)}$ and $g_j^{(\epsilon)}$ via the marginal conditions:

$$(\Gamma\mathbf{1})_i = \hat{\nu}_{src,i} e^{f_i^{(\epsilon)}/\lambda_{src}}, \quad (\Gamma^\top\mathbf{1})_j = \hat{\nu}_{tar,j} e^{g_j^{(\epsilon)}/\lambda_{tar}}. \tag{33}$$

Substituting these into the optimality condition, the optimal coupling takes the Gibbs form:

$$\Gamma^*_{ij}(\epsilon) = \exp\left(-\frac{C_{ij} + f_i^{(\epsilon)} + g_j^{(\epsilon)}}{\epsilon}\right). \tag{34}$$

Summing over $j$ to recover the source marginal yields the self-consistent equation:

$$\hat{\nu}_{\text{src},i} e^{f_i^{(\epsilon)}/\lambda_{\text{src}}} = \sum_j \exp\left(-\frac{C_{ij} + f_i^{(\epsilon)} + g_j^{(\epsilon)}}{\epsilon}\right). \tag{35}$$

Taking the logarithm and rearranging terms to isolate $f_i^{(\epsilon)}$:

$$f_i^{(\epsilon)}\left(\frac{1}{\lambda_{\text{src}}} + \frac{1}{\epsilon}\right) = \log \sum_j \exp\left(-\frac{C_{ij} + g_j^{(\epsilon)}}{\epsilon}\right) - \log \hat{\nu}_{\text{src},i}. \tag{36}$$

Multiplying by $\epsilon$:

$$f_i^{(\epsilon)}\left(\frac{\epsilon}{\lambda_{\text{src}}} + 1\right) = \epsilon \log \sum_j \exp\left(-\frac{C_{ij} + g_j^{(\epsilon)}}{\epsilon}\right) - \epsilon \log \hat{\nu}_{\text{src},i}. \tag{37}$$

We now take the limit $\epsilon \to 0$. We invoke the convergence of regularized potentials to their unregularized counterparts $(f^{(0)}, g^{(0)})$ [Peyré & Cuturi, 2019]. Applying the Laplace principle (Log-Sum-Exp limit) to the RHS:

$$\lim_{\epsilon \to 0} \epsilon \log \sum_j \exp\left(\frac{-(C_{ij} + g_j^{(\epsilon)})}{\epsilon}\right) = \max_j \left[-(C_{ij} + g_j^{(0)})\right] = -\min_j(C_{ij} + g_j^{(0)}). \tag{38}$$

Since $\epsilon/\lambda_{\text{src}} \to 0$ and the constant term $\epsilon \log \hat{\nu} \to 0$, Eq. (37) converges to the stated result in Eq. (31). $\qquad\square$

With the asymptotic behavior established, we now complete the proof of the main proposition.

***Proof of Proposition 3.2.*** From the definition of the dual potentials in Eq. (33), the optimal relevance weight $w_i^*$ is given by:

$$w_i^* = (\Gamma^* \mathbf{1})_i = \hat{\nu}_{\text{src},i} \exp\left(\frac{f_i^{(\epsilon)}}{\lambda_{\text{src}}}\right). \tag{39}$$

Substituting the asymptotic limit from Lemma C.2, we obtain the behavior as $\epsilon \to 0$:

$$\frac{w_i^*}{\hat{\nu}_{\text{src},i}} \approx \exp\left(-\frac{\min_j(C_{ij} + g_j^{(0)})}{\lambda_{\text{src}}}\right). \tag{40}$$

Under Assumption C.1, the domain is compact and the cost function is bounded. In the unregularized limit, the target potential $g^{(0)}$ is the $c$-transform of $f^{(0)}$ adjusted for marginal penalties, which ensures it remains bounded within the compact support. Specifically, there exists a constant $B_{\text{tar}} \geq 0$ such that $|g_j^{(0)}| \leq B_{\text{tar}}$ for all $j \in \{1, \ldots, N\}$.

Let $C_{\min}(x_i) = \min_j C_{ij}$ denote the minimum geometric distance from source sample $x_i$ to the target support. We can bound the effective transport cost term in Eq. (40):

$$C_{\min}(x_i) - B_{\text{tar}} \leq \min_j(C_{ij} + g_j^{(0)}) \leq C_{\min}(x_i) + B_{\text{tar}}. \tag{41}$$

Since the exponential function $e^{-x/\lambda}$ is monotonically decreasing, substituting these bounds into Eq. (40) yields:

$$\exp\left(-\frac{C_{\min}(x_i) + B_{\text{tar}}}{\lambda_{\text{src}}}\right) \leq \frac{w_i^*}{\hat{\nu}_{\text{src},i}} \leq \exp\left(-\frac{C_{\min}(x_i) - B_{\text{tar}}}{\lambda_{\text{src}}}\right). \tag{42}$$

This confirms that the weights decay exponentially with respect to the geometric distance $C_{\min}(x_i)$, modulo a bounded constant factor $B_{\text{tar}}$ determined by the target geometry. $\qquad\square$

# D. Environment Setting

To ensure fair comparison and reproducibility, we follow the cross-domain evaluation protocols established in OTDF (Lyu et al., 2025) and ODRL (Lyu et al., 2024b). Our experiments involve two complementary environment settings. First, for the main cross-domain offline RL experiments in Section 5.1 and the parameter studies in Section 5.3, we follow the OTDF-style benchmark, which evaluates policy adaptation across gravity, kinematic, and morphology shifts using scarce target-domain datasets. Second, for the ODRL comparison in Section 5.2, we follow the ODRL protocol (Lyu et al., 2024b), which evaluates off-dynamics offline RL under gravity and friction shifts with multiple shift levels.

We adopt four standard MuJoCo continuous control tasks: HalfCheetah, Hopper, Walker2d, and Ant. The source domain corresponds to the vanilla MuJoCo environment, while the target domain is generated by modifying the physical parameters of the environment. All settings preserve the state space, action space, and reward function, and only modify the transition dynamics through changes in physical properties.

## D.1. OTDF-Style Cross-Domain Offline RL Setting

This setting is used for the main results in Section 5.1 and the parameter studies in Section 5.3. It focuses on source-to-target policy adaptation under scarce target-domain data and includes three types of dynamics shifts: gravity, kinematic, and morphology shifts.

**Datasets.**   We use the standard D4RL MuJoCo datasets (Fu et al., 2020) as source-domain datasets. Following the OTDF protocol (Lyu et al., 2025), target-domain datasets are collected in the modified target environments using SAC-trained policies (Haarnoja et al., 2018). To simulate realistic data scarcity, each target-domain dataset contains only 5 trajectories, corresponding to approximately 5,000 transitions. We consider three target data qualities:

- *medium*: trajectories from a medium-quality target-domain policy;

- *expert*: trajectories from an expert target-domain policy;

- *medium-expert*: a mixture of 2 medium and 3 expert trajectories.

**Gravity shift.**   The gravity shift simulates changes in the global gravitational acceleration acting on the robot. In the OTDF-style setting, the target-domain gravity is reduced relative to the source domain:

```
<option gravity="0 0 -4.905" timestep="0.01"/>
```

*Listing 1.* XML modification for the OTDF-style gravity shift.

This corresponds to changing the gravitational acceleration from the standard MuJoCo value $g = -9.8$ to $g = -4.905$, while keeping the remaining environment components unchanged.

**Kinematic shift.**   The kinematic shift simulates mechanical failures by restricting the rotation ranges of selected joints. These modifications change the feasible motion patterns of the robot without changing the state or action spaces. The adopted kinematic modifications include:

- **HalfCheetah:** the range of the back thigh joint is severely restricted, making part of the hind leg effectively broken;

- **Hopper:** the ranges of selected head and foot joints are reduced;

- **Walker2d:** the foot joint range is restricted, limiting normal walking motion;

- **Ant:** selected hip joints are constrained, impairing multidirectional locomotion.

An example XML modification for HalfCheetah is:

```
<joint axis="0 1 0" name="bthigh" pos="0 0 0"
range="-.0052 .0105" type="hinge"/>
```

*Listing 2.* Example XML modification for the OTDF-style kinematic shift in HalfCheetah.

**Morphology shift.** The morphology shift simulates structural changes in the robot body by modifying the sizes of selected limbs or body parts. Unlike the kinematic shift, which restricts joint motion ranges, the morphology shift changes the robot geometry while keeping the state and action spaces fixed. The adopted morphology modifications include:

- **HalfCheetah:** the sizes of the back and front thighs are modified;

- **Hopper:** the head size is enlarged;

- **Walker2d:** selected thigh and foot components are resized, creating asymmetric body geometry;

- **Ant:** selected feet are shrunk, reducing stability during locomotion.

An example XML modification for Ant is:

```
1   <geom name="left_ankle_geom" size="0.08" type="capsule"/>
2   <geom name="right_ankle_geom" size="0.08" type="capsule"/>
```

*Listing 3.* Example XML modification for the OTDF-style morphology shift in Ant.

### D.2. ODRL/MOBODY Gravity-Friction Setting

This setting is used for the comparison with recent off-dynamics offline RL baselines in Section 5.2. Different from the OTDF-style setting above, this benchmark follows the ODRL locomotion protocol (Lyu et al., 2024b). The purpose of this setting is to evaluate whether ROOT remains competitive under systematically varied dynamics-shift levels. The source domain is the vanilla MuJoCo environment, and the target domain is obtained by modifying one physical property of the source environment. In Section 5.2, we focus on two types of dynamics shifts: gravity and friction and use medium-quality offline datasets for both domains.

**Gravity shift.** In the ODRL gravity-shift setting, only the magnitude of the gravitational acceleration is changed, while the direction of gravity remains fixed. Let $g_{\mathrm{src}}$ denote the source-domain gravity. The target-domain gravity is defined as $g_{\mathrm{tar}} = \alpha_g g_{\mathrm{src}}$, where the shift level $\alpha_g$ is chosen from $\alpha_g \in \{0.1,\ 0.5,\ 2.0,\ 5.0\}$. Thus, values smaller than 1.0 correspond to weaker gravity than the source domain, while values larger than 1.0 correspond to stronger gravity. The corresponding ODRL task names follow the format: `[env]-gravity-[level]`, for example, `halfcheetah-gravity-0.5` denotes a HalfCheetah target domain whose gravity magnitude is 0.5 times that of the source domain.

**Friction shift.** In the ODRL friction-shift setting, the friction attributes of MuJoCo contacts are modified. MuJoCo represents friction using multiple components, including sliding, torsional, and rolling friction. Following ODRL (Lyu et al., 2024b), all friction components are scaled by the same factor. Let $f_{\mathrm{src}}$ denote the source-domain friction vector. The target-domain friction is defined as $f_{\mathrm{tar}} = \alpha_f f_{\mathrm{src}}$, where $\alpha_f \in \{0.1,\ 0.5,\ 2.0,\ 5.0\}$. This produces target domains with either substantially reduced or increased contact friction. The corresponding ODRL task names follow the format: `[env]-friction-[level]`, for example, `hopper-friction-2.0` denotes a Hopper target domain whose friction components are 2.0 times those of the source domain.

### D.3. Reference Scores

In this work, we use the standard normalized score metric proposed by D4RL. The normalized score is calculated as $100 \times \frac{\text{score}-\text{random\_score}}{\text{expert\_score}-\text{random\_score}}$. Table 3 lists the reference minimum (random) and maximum (expert) scores for the environments used in my experiments.

## E. Implementation Details

Our framework is implemented using **PyTorch** for the offline RL backbone and **JAX** with the **OTT (Optimal Transport Tools)** library (Cuturi et al., 2022) for the high-performance UOT solver. All experiments were conducted on a single NVIDIA GeForce RTX A6000 GPU.

*Table 3.* Reference scores for D4RL MuJoCo environments used in normalization.

| Environment | Dynamics shift Type | Random Score | Expert Score |
|---|---|---|---|
| HalfCheetah | Gravity | -280.18 | 9509.15 |
| HalfCheetah | Kinematic | -280.18 | 7065.03 |
| HalfCheetah | Morphology | -280.18 | 9713.59 |
| Hopper | Gravity | -26.336 | 3234.3 |
| Hopper | Kinematic | -26.336 | 2842.73 |
| Hopper | Morphology | -26.336 | 3152.75 |
| Walker2d | Gravity | 10.08 | 5194.713 |
| Walker2d | Kinematic | 10.08 | 3257.51 |
| Walker2d | Morphology | 10.08 | 4398.43 |
| Ant | Gravity | -325.6 | 4317.065 |
| Ant | Kinematic | -325.6 | 5122.57 |
| Ant | Morphology | -325.6 | 5722.01 |

### E.1. Scalable UOT Solver

To avoid solving repetitive high-dimensional optimization problems during the policy training loop, we decouple the geometric alignment phase from the policy learning phase. We solve the Unbalanced Optimal Transport (UOT) problem as a pre-processing step, computing a static set of relevance weights $w^*$ for the entire source dataset. This design leverages the efficiency of the OTT-JAX library (Cuturi et al., 2022) to handle large-scale datasets (e.g., 1M transitions in D4RL) within minutes.

*Table 4.* Hyperparameters for ROOT.

| Hyperparameter | Value |
|---|---|
| Batch Size (Block-wise) | 10,000 |
| Entropic Reg ($\epsilon$) | 0.01 |
| Source Relaxation ($\lambda_{\text{src}}$) | 0.05 |
| Target Relaxation ($\lambda_{\text{tar}}$) | 0.5 |
| Discount Factor ($\gamma$) | 0.99 |
| Target Smoothing ($\tau$) | 0.005 |
| Expectile ($\tau_{\text{IQL}}$) | 0.7 |
| Temperature ($\beta$) | 3.0 |
| Target Behavior Constraint Strength ($\lambda_\beta$) | 0.5 |
| Actor Learning Rate | $3 \times 10^{-4}$ |
| Critic Learning Rate | $3 \times 10^{-4}$ |
| Batch Size | 256 |
| Hidden Dim (Actor/Critic) | 256 |
| Hidden Dim (CVAE) | 750 |

**Geometry and Block-wise Computation.** We define the transport geometry using the Euclidean cost function on standardized state-action features. Specifically, we normalize both source and target datasets by subtracting their joint mean and dividing by the joint standard deviation to ensure numerical stability. To overcome GPU memory constraints when computing the $M \times N$ cost matrix, we implement a **block-wise Sinkhorn solver**. The source dataset is partitioned into mini-batches (e.g., size $10,000$), and the UOT problem is solved between each batch and the full target dataset. We utilize `jax.jit` to just-in-time compile the Sinkhorn fixed-point iterations, enabling rapid convergence.

**Zero-Copy Data Pipeline.** A critical engineering challenge is the interoperability between the PyTorch-based RL pipeline and the JAX-based solver. We implement a zero-copy data transfer mechanism using **DLPack**. This protocol allows memory pointers to be passed directly between PyTorch tensors and JAX arrays on the GPU, eliminating the overhead of CPU-GPU synchronization and redundant data copying.

## E.2. Offline RL Architecture

We build ROOT on top of the Implicit Q-Learning (IQL) (Kostrikov et al., 2022) algorithm. We modify the standard IQL update rules to incorporate the geometry-aware weights $w^*$ and target support constraint.

**Network Architecture.** Following standard practices, the actor $\pi_\phi(a|s)$ and the twin critics $Q_{\theta_1}, Q_{\theta_2}$ are parametrized as Multi-Layer Perceptrons (MLPs) with 2 hidden layers of 256 units and ReLU activations. The actor outputs a Gaussian distribution with state-dependent mean and diagonal covariance. To enforce the target support constraint, we model the target behavior policy $\pi_\beta(a|s)$ using a **Conditional Variational Autoencoder (CVAE)**. The CVAE consists of an encoder and a decoder, each containing two intermediate layers with **750 hidden units** and ReLU activations. The latent dimension is set to twice the action dimension $(2 \times \dim(\mathcal{A}))$.

**Weighted Objective Implementation.** The raw relevance weights $w^*$ obtained from the UOT solver are first normalized by their global mean and then clipped to $[0, 5.0]$ for numerical stability. Threshold-based filtering with $\tau_{\text{th}}$ is used only in the ablation study and is not part of the default ROOT pipeline.

$$w_{\text{final}} = \text{clip}\left(w^*, 0.0, 5.0\right). \tag{43}$$

These processed weights are then used to reweight the mean squared Bellman error for the critic and the advantage-weighted regression loss for the actor, effectively steering the policy learning towards the target-aligned manifold.

## E.3. Hyperparameters

Table 4 lists the common hyperparameters used across all tasks. We use the Adam optimizer for all networks.

# F. Extended Experimental Results

In this section, we provide additional empirical results that were omitted from the main text due to space constraints. This includes extensive results on gravity and morphology shifts, extended results on halfcheetah-friction and halfcheetah-kinematic-footjnt setting with different shift levels.

## F.1. Full Benchmark Results on Gravity and Morphology Shifts

In this section, we provide the complete numerical results for the **Gravity Shift** and **Morphology Shift** benchmarks, corresponding to the aggregated results discussed in Section 5.1.

**Analysis of Gravity Shift (Table 5).** Gravity shift introduces a global parameter change that affects the entire state space uniformly. As shown in Table 5, ROOT achieves dominant performance in **Hopper** and **Walker2d** tasks, surpassing the strong baseline DmC by a significant margin (e.g., **64.7** vs. 51.3 in *Hopper-m-expert*). This suggests that for environments with complex balance constraints, our UOT-based alignment better preserves the functional manifold than local matching methods. We observe that DmC performs exception well on *Ant* tasks under gravity shift. This is likely because the stable, four-legged structure of the Ant robot makes it robust to simple gravity scaling, allowing local heuristic matching to find shortcuts. However, ROOT remains the second-best performer and consistently outperforms the global OT baseline (OTDF), validating the benefit of relaxing the mass conservation constraint.

**Analysis of Morphology Shift (Table 6).** Morphology shift represents the most challenging scenario, where the agent's physical structure (e.g., limb size) changes, creating a disjoint support between domains. Table 6 demonstrates that ROOT achieves the highest **Total Score (1285.8)**, outperforming OTDF (1274.3) and significantly surpassing other baselines like IQL (798.0) and DARA (816.8). Notably, in *HalfCheetah* and *Hopper*, ROOT secures the best score in almost every setting. The comparison with OTDF is particularly illuminating here: OTDF forces a mapping between the source limb and the modified target limb, which can be detrimental when the structural mismatch is severe. In contrast, ROOT's *transport-or-discard* mechanism allows the agent to ignore source transitions that physically conflict with the new morphology, leading to safer and more stable transfer.

## F.2. Time and Memory Analysis Compared with OTDF

To further examine the computational cost of ROOT, we compare its time and memory consumption with OTDF during both the weight-computation stage and the downstream policy-training stage. All measurements are conducted under the same

*Table 5.* **Performance comparison in Gravity Shift tasks.** We report normalized scores and standard deviations over 5 seeds. Source abbreviations use the form Env-Type: Ant/Half/Hopp/Walk denote Ant, HalfCheetah, Hopper, and Walker2d, respectively; m, me, and mr denote medium, medium-expert, and medium-replay source datasets. The Target column denotes the target-domain dataset quality. The best score in each setting is highlighted in bold.

| Source | Target | IQL | DARA | BOSA | SRPO | IGDF | OTDF | DmC | **ROOT** |
|---|---|---|---|---|---|---|---|---|---|
| Ant-m | medium | 10.2±1.8 | 9.4±0.9 | 12.4±2.0 | 11.7±1.0 | 11.3±1.3 | 45.1±12.4 | **56.9±2.2** | 27.0±3.0 |
| Ant-m | med-expert | 9.4±1.2 | 10.0±0.9 | 11.6±1.3 | 10.2±1.2 | 9.4±1.4 | 33.9±5.4 | **47.5±3.9** | 17.4±2.4 |
| Ant-m | expert | 10.2±0.3 | 9.8±0.6 | 11.8±0.4 | 9.5±0.6 | 9.7±1.6 | 33.2±9.0 | **36.1±7.8** | 20.7±1.9 |
| Ant-me | medium | 9.8±2.4 | 8.1±1.8 | 8.1±3.0 | 8.4±2.1 | 8.9±1.5 | 18.6±11.9 | **55.7±8.1** | 28.4±6.0 |
| Ant-me | med-expert | 9.0±0.8 | 6.4±1.4 | 6.2±1.5 | 6.1±3.5 | 7.2±2.9 | 34.0±9.4 | **46.3±4.9** | 17.6±0.8 |
| Ant-me | expert | 9.1±2.6 | 10.4±2.9 | 4.2±3.9 | 8.8±1.0 | 9.2±1.5 | 23.2±2.9 | **42.7±13.0** | 20.9±4.2 |
| Ant-mr | medium | 18.9±2.6 | 21.7±2.1 | 13.9±1.5 | 18.7±1.7 | 19.6±1.0 | 29.6±10.7 | **41.8±4.7** | 31.4±1.1 |
| Ant-mr | med-expert | 19.1±3.0 | 18.3±2.1 | 15.9±2.7 | 18.7±1.8 | 20.3±1.6 | 25.4±2.1 | 27.6±0.1 | **28.5±1.8** |
| Ant-mr | expert | 18.5±0.9 | 20.0±1.3 | 14.5±1.7 | 19.9±2.1 | 18.8±2.1 | 24.5±2.8 | 28.0±0.3 | **28.3±1.5** |
| Half-m | medium | 39.6±3.3 | 41.2±3.9 | 38.9±4.0 | 36.9±4.5 | 36.6±5.5 | 40.7±7.7 | **48.0±0.6** | 47.7±0.9 |
| Half-m | med-expert | 39.6±3.7 | 40.7±2.8 | 40.4±3.0 | 40.7±2.3 | 38.7±6.2 | 28.6±3.2 | **48.9±0.7** | 48.6±0.8 |
| Half-m | expert | 42.4±3.8 | 39.8±4.4 | 40.5±3.9 | 39.4±1.6 | 39.6±4.6 | 36.1±5.3 | **48.8±1.0** | 48.4±1.1 |
| Half-me | medium | 38.6±6.0 | 37.8±3.3 | 41.8±5.1 | 42.5±2.3 | 37.7±7.3 | 35.9±4.5 | **49.9±1.4** | 49.0±1.7 |
| Half-me | med-expert | 39.6±3.0 | 39.4±4.4 | 38.7±3.7 | 43.3±2.7 | 40.7±3.2 | 32.4±5.5 | 48.1±1.9 | **48.3±1.3** |
| Half-me | expert | 43.4±0.9 | 45.3±1.3 | 39.9±2.7 | 43.3±3.0 | 41.1±4.1 | 26.5±9.1 | **51.0±0.8** | 48.9±0.8 |
| Half-mr | medium | 20.1±5.0 | 17.6±6.2 | 20.0±4.9 | 17.5±5.2 | 14.4±2.2 | 21.5±6.5 | 32.2±0.8 | **35.7±2.1** |
| Half-mr | med-expert | 17.2±1.6 | 20.2±5.2 | 16.7±4.2 | 16.3±1.7 | 10.0±2.5 | 14.7±4.1 | 30.3±5.1 | **34.0±1.6** |
| Half-mr | expert | 20.7±5.5 | 22.4±1.7 | 15.4±4.2 | 23.1±4.0 | 15.3±3.7 | 11.4±1.9 | 32.7±2.6 | **34.9±3.4** |
| Hopp-m | medium | 11.2±1.1 | 17.3±3.8 | 15.2±3.3 | 12.4±1.0 | 15.3±3.5 | 32.4±8.0 | 30.6±5.5 | **50.3±5.4** |
| Hopp-m | med-expert | 14.7±3.6 | 15.4±2.5 | 21.1±9.3 | 14.2±1.8 | 15.1±3.6 | 24.2±3.6 | 35.7±1.4 | **52.5±5.5** |
| Hopp-m | expert | 12.5±1.6 | 19.3±10.5 | 12.7±1.7 | 11.8±0.9 | 14.8±4.0 | 33.7±7.8 | 51.3±10.6 | **55.4±13.9** |
| Hopp-me | medium | 19.1±6.6 | 18.5±12.3 | 15.9±5.9 | 19.7±8.5 | 22.3±5.4 | 26.4±10.1 | 52.2±6.6 | **65.9±12.3** |
| Hopp-me | med-expert | 16.8±2.7 | 16.0±6.1 | 17.3±2.5 | 15.8±3.3 | 16.6±7.7 | 28.3±6.7 | **50.4±11.3** | 49.8±7.9 |
| Hopp-me | expert | 20.9±4.1 | 23.9±14.8 | 23.2±7.9 | 21.4±1.9 | 26.0±9.2 | 44.9±10.6 | 52.8±10.3 | **60.3±10.6** |
| Hopp-mr | medium | 13.9±2.9 | 10.7±4.3 | 3.3±1.9 | 14.0±2.6 | 15.3±4.4 | 31.1±13.4 | 35.4±0.7 | **35.8±2.3** |
| Hopp-mr | med-expert | 13.3±6.3 | 12.5±5.6 | 4.6±1.7 | 14.4±4.2 | 15.4±5.5 | 24.2±6.1 | **41.0±2.0** | 40.4±3.5 |
| Hopp-mr | expert | 11.0±2.6 | 14.3±6.0 | 3.2±0.8 | 16.4±5.0 | 16.1±4.0 | 31.0±9.8 | 29.9±7.8 | **34.0±5.3** |
| Walk-m | medium | 28.1±12.9 | 28.4±13.7 | 38.0±11.2 | 21.4±7.0 | 22.1±8.4 | 36.6±2.3 | 52.5±2.0 | **54.8±3.0** |
| Walk-m | med-expert | 35.7±4.7 | 30.7±9.7 | 40.9±7.2 | 34.0±9.9 | 35.4±9.1 | 44.8±7.5 | 59.2±2.7 | **60.1±2.5** |
| Walk-m | expert | 37.3±8.0 | 36.0±7.0 | 41.3±8.6 | 39.5±3.8 | 36.2±13.6 | 44.0±4.0 | **63.8±2.7** | 63.0±2.7 |
| Walk-me | medium | 39.9±13.1 | 41.6±13.0 | 32.3±7.2 | 46.4±3.5 | 33.8±3.1 | 30.2±9.8 | 57.5±3.3 | **58.5±6.5** |
| Walk-me | med-expert | 49.1±6.9 | 45.8±9.4 | 40.1±4.5 | 36.4±3.4 | 44.7±2.9 | 53.3±7.1 | **67.8±4.0** | 62.5±2.8 |
| Walk-me | expert | 40.4±11.9 | 56.4±3.5 | 43.7±4.4 | 45.8±8.0 | 45.3±10.4 | 61.1±3.4 | **67.1±4.8** | 65.7±3.3 |
| Walk-mr | medium | 14.6±2.5 | 14.1±6.1 | 7.6±5.8 | 17.9±3.8 | 11.6±4.6 | 32.7±7.0 | **42.2±5.9** | 40.3±4.5 |
| Walk-mr | med-expert | 15.3±1.9 | 15.9±5.8 | 4.8±5.8 | 15.3±4.5 | 13.9±6.5 | 31.6±6.1 | 31.5±5.2 | **41.1±6.5** |
| Walk-mr | expert | 15.8±7.2 | 15.7±4.5 | 7.1±4.6 | 13.7±8.1 | 15.2±5.3 | 31.3±5.3 | 39.3±6.2 | **40.9±2.0** |
| **Total Score** | | 825.0 | 851.0 | 763.2 | 825.5 | 803.6 | 1160.7 | **1632.7** | 1547.0 |

experimental setting and hardware configuration. The results are summarized in Table 7.

As shown in Table 7, ROOT requires substantially less memory during the pre-training weight-computation stage. Compared with OTDF, ROOT reduces pre-training memory consumption from 44290 MiB to 31356 MiB, corresponding to a reduction of approximately 29.2%. This is because ROOT only needs the source marginal of the transport plan to derive sample-wise relevance weights, whereas OTDF relies on a dense transport-based matching procedure to compute deviation scores.

The pre-training time of ROOT is longer than that of OTDF. This difference is mainly caused by the more conservative Sinkhorn configuration used in ROOT: we adopt a larger maximum iteration limit to ensure stable convergence of the robust unbalanced OT objective, while OTDF uses a smaller iteration budget. Importantly, this additional cost is incurred only once before policy learning. During downstream training, ROOT introduces only a small overhead: the training time increases from 2412 s to 2524 s per 1M steps, and the training memory increases from 1714 MiB to 1986 MiB. These results indicate that ROOT trades a moderate one-time pre-training cost for lower pre-training memory consumption and more adaptive source-transition weighting.

### F.3. Detailed Analysis of Relevance Weight Distributions

To provide a microscopic view of how ROOT achieves robust source data filtering compared to standard balanced Optimal Transport methods (e.g., OTDF), we visualize and analyze the distribution of the learned relevance weights $w^*$.

**Visualization Setup** We use hopper environment with kinematic shift, where source domain is medium quality and target domain is expert quality. This scenario represents a significant quality gap where a large portion of the source data

*Table 6.* **Performance comparison in Morphology Shift tasks.** We report normalized scores and standard deviations over 5 seeds. Morphology shifts modify robot morphology (e.g., limb size or joints), creating substantial support mismatch. Source abbreviations use the form Env-Type: Ant/Half/Hopp/Walk denote Ant, HalfCheetah, Hopper, and Walker2d, respectively; m, me, and mr denote medium, medium-expert, and medium-replay source datasets. The Target column denotes the target-domain dataset quality. The best score in each setting is highlighted in bold.

| Source | Target | IQL | DARA | BOSA | SRPO | IGDF | OTDF | **ROOT** |
|---|---|---|---|---|---|---|---|---|
| Half-m | medium | 30.0±1.6 | 26.6±3.3 | 19.3±3.5 | 41.3±0.4 | 41.6±0.5 | 39.1±2.3 | **43.0±0.1** |
| Half-m | med-expert | 31.8±1.1 | 32.0±0.7 | 33.6±1.1 | 30.7±0.8 | 29.6±2.2 | **35.6±0.7** | 35.5±1.1 |
| Half-m | expert | 8.5±1.0 | 9.3±1.6 | 7.9±0.8 | 8.6±0.9 | 10.0±0.8 | 10.7±1.2 | **11.4±0.5** |
| Half-mr | medium | 30.8±4.4 | 35.6±0.7 | 35.0±4.6 | 32.0±1.4 | 28.0±2.0 | 40.0±1.2 | **41.3±0.2** |
| Half-mr | med-expert | 12.9±2.2 | 16.9±4.1 | 19.9±5.5 | 12.4±1.6 | 12.0±3.7 | **34.4±0.7** | 30.4±1.7 |
| Half-mr | expert | 5.9±1.7 | 3.7±2.7 | 2.4±1.9 | 6.2±1.4 | 5.3±2.3 | 8.2±2.7 | **8.7±2.0** |
| Half-me | medium | 41.5±0.1 | 40.3±1.2 | 41.3±0.3 | 41.3±0.4 | 40.9±0.4 | 41.4±0.3 | **42.7±0.2** |
| Half-me | med-expert | 25.8±2.0 | 30.6±2.8 | 32.1±0.8 | 27.2±0.8 | 26.2±1.8 | **35.1±0.6** | 33.0±0.5 |
| Half-me | expert | 7.8±1.3 | 8.3±1.3 | 9.1±0.8 | 7.8±0.9 | 7.5±0.9 | 9.8±1.0 | **10.2±1.9** |
| Hopp-m | medium | **13.5±0.2** | **13.5±0.4** | 13.2±0.3 | 13.4±0.1 | 13.4±0.2 | 11.0±0.9 | **13.5±0.6** |
| Hopp-m | med-expert | 13.4±0.1 | 13.6±0.2 | 11.2±4.6 | 13.3±0.2 | 13.3±0.4 | 12.6±0.8 | **14.0±0.3** |
| Hopp-m | expert | 13.5±0.2 | 13.6±0.3 | 13.3±0.4 | 13.6±0.2 | **13.9±0.1** | 10.7±4.7 | 13.7±0.5 |
| Hopp-mr | medium | 10.8±1.1 | 10.2±1.0 | 1.2±0.0 | 10.7±1.6 | 12.0±4.4 | 8.7±2.8 | **17.3±5.9** |
| Hopp-mr | med-expert | 11.6±1.6 | 10.4±0.9 | 1.3±0.2 | 10.4±1.2 | 8.2±2.8 | 9.7±2.7 | **19.1±7.5** |
| Hopp-mr | expert | 9.8±0.5 | 9.0±0.3 | 1.3±0.1 | 10.4±1.4 | 11.4±1.5 | 10.7±2.4 | **14.5±5.2** |
| Hopp-me | medium | 12.6±1.4 | 13.0±0.5 | 15.7±7.2 | 14.0±2.3 | 12.7±0.8 | 7.9±3.2 | **20.4±5.6** |
| Hopp-me | med-expert | 14.1±1.3 | 13.8±0.6 | 12.0±1.4 | 13.5±0.3 | 13.3±1.2 | 9.6±3.5 | **17.4±1.6** |
| Hopp-me | expert | 13.8±0.5 | 12.3±1.8 | 10.5±5.0 | 14.7±2.3 | 12.8±0.9 | 5.9±4.0 | **14.9±1.1** |
| Walk-m | medium | 23.0±4.7 | 23.3±3.3 | 6.2±2.9 | 24.7±1.7 | 27.5±9.5 | 50.5±5.8 | **57.3±4.1** |
| Walk-m | med-expert | 21.5±8.6 | 22.2±7.6 | 7.2±2.9 | 18.7±7.3 | 20.7±5.9 | 44.3±23.8 | **48.1±9.2** |
| Walk-m | expert | 20.3±2.8 | 17.3±3.4 | 15.8±8.7 | 21.1±7.2 | 15.8±4.5 | 55.3±8.3 | **58.3±10.7** |
| Walk-mr | medium | 11.3±3.0 | 10.9±4.6 | 5.4±4.0 | 10.4±4.8 | 13.4±7.2 | 37.4±5.1 | **40.5±3.9** |
| Walk-mr | med-expert | 7.0±1.5 | 4.5±1.1 | 4.0±2.2 | 4.9±1.7 | 6.9±2.2 | **33.8±6.9** | 24.5±3.1 |
| Walk-mr | expert | 6.3±0.9 | 4.5±1.1 | 3.8±3.4 | 5.5±0.9 | 5.5±2.2 | **41.5±6.8** | 15.5±1.4 |
| Walk-me | medium | 24.1±7.4 | 31.7±6.6 | 18.7±6.5 | 29.9±4.7 | 27.5±2.3 | 49.9±4.6 | **52.7±2.8** |
| Walk-me | med-expert | 27.0±5.5 | 23.3±5.5 | 11.1±0.9 | 22.9±3.8 | 25.3±6.4 | 40.5±11.0 | **49.6±5.8** |
| Walk-me | expert | 22.4±3.3 | 25.2±5.7 | 9.9±3.9 | 18.7±5.7 | 24.7±2.4 | 45.7±6.9 | **58.8±9.1** |
| Ant-m | medium | 38.7±3.8 | 41.3±1.8 | 18.2±1.9 | 40.6±2.1 | 40.9±1.7 | 39.4±1.7 | **44.0±0.3** |
| Ant-m | med-expert | 47.0±5.1 | 43.3±2.0 | 45.3±7.0 | 47.2±4.3 | 44.4±1.7 | **58.3±8.9** | 55.3±3.2 |
| Ant-m | expert | 36.2±3.5 | 48.5±4.2 | 72.2±10.5 | 42.2±9.9 | 41.4±4.2 | **85.4±4.4** | 67.7±9.1 |
| Ant-mr | medium | 38.2±2.9 | 38.9±2.7 | 20.2±3.7 | 38.3±1.9 | 39.7±1.2 | 41.2±0.9 | **43.4±0.3** |
| Ant-mr | med-expert | 38.1±3.5 | 33.4±5.5 | 15.2±1.6 | 35.0±5.7 | 37.3±2.4 | 50.8±4.5 | **52.3±2.3** |
| Ant-mr | expert | 24.1±1.9 | 24.5±2.6 | 16.0±1.7 | 22.7±3.0 | 23.6±1.4 | **67.2±7.5** | 44.5±4.9 |
| Ant-me | medium | 32.9±5.1 | 40.2±1.5 | 28.1±5.6 | 35.9±2.5 | 36.1±4.4 | 39.9±2.9 | **44.0±0.3** |
| Ant-me | med-expert | 35.7±3.9 | 36.5±8.7 | 14.8±15.9 | 24.5±15.7 | 30.7±10.8 | **65.7±4.5** | 58.8±3.9 |
| Ant-me | expert | 36.1±8.5 | 34.6±5.8 | 53.9±5.0 | 38.4±9.4 | 35.2±6.6 | **86.4± 2.2** | 67.3±9.9 |
| **Total Score** | | 798.0 | 816.8 | 646.3 | 803.1 | 808.7 | 1274.3 | **1283.6** |

*Table 7.* Time and memory comparison between OTDF and ROOT. "Pre-train" denotes the source-weight computation stage before downstream offline policy learning. "Train" denotes the downstream policy-training stage for 1M gradient steps.

| Method | Pre-train Mem | Pre-train Time | Train Mem | Train Time |
|---|---|---|---|---|
| OTDF | 44290 MiB | 26.99 s | 1714 MiB | 2412 s / 1M steps |
| ROOT | 31356 MiB | 75.85 s | 1986 MiB | 2524 s / 1M steps |

(medium-quality transitions) is likely sub-optimal or irrelevant for recovering the expert policy, making it an ideal testbed for outlier rejection capabilities. Further to ensure a fair comparison reflecting the actual signals used for policy updates, we visualize the weights after their respective post-processing steps.

**Natural Sparsity vs. Forced Participation (Left Panel).** The log-scale histogram reveals that ROOT's weight distribution (Orange) is highly bimodal. It features a massive density peak near zero ($w \approx 10^{-4}$) and a long tail extending to high values ($w > 4$). This empirically validates **Proposition 3.2**, confirming that ROOT naturally suppresses kinematically inconsistent transitions via exponential decay.

In stark contrast, OTDF (Blue) exhibits a "box-like" distribution concentrated strictly within the interval $[e^{-1}, 1] \approx [0.36, 1.0]$. This bounds implies that even the worst source sample is assigned at least 36% of the maximum weight. This is a direct consequence of the *mass conservation constraint* in standard OT, which forces the algorithm to distribute probability mass "quotas" across all samples, preventing true outlier rejection.

**Self-Adaptive Filtering vs. Dependency on Heuristics (Right Panel).** The sorted spectrum (Lorenz curve) further clarifies

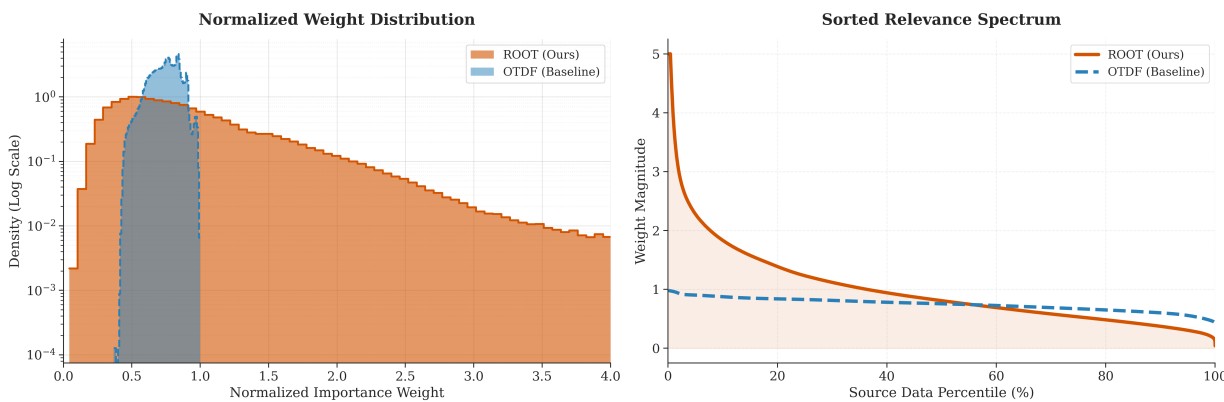

*Figure 4.* **Comparison of Relevance Weight Distributions on Hopper-Kinematic Medium → Expert. (Left)** The histogram (log scale) shows that ROOT (orange) naturally pushes the majority of weights towards zero (auto-rejection), while OTDF (blue) maintains a dense "wall" of non-negligible weights, illustrating the forced matching phenomenon. **(Right)** The sorted relevance spectrum demonstrates ROOT's heavy-tailed sparsity, whereas OTDF exhibits a flat plateau requiring manual hard truncation.

the implication for policy learning. ROOT's curve (Orange) shows a sharp "L-shaped" decay, effectively implementing a *soft subset selection* mechanism where only the top percentile of data contributes to the gradient. Crucially, the "knee" of this curve is determined adaptively by the UOT formulation based on geometric consensus, not by a fixed hyperparameter.

Conversely, the OTDF curve (Blue dashed) is nearly linear and flat. Without the artificial Top-$K$ truncation (filtering threshold $\tau_{\mathrm{th}}$), OTDF would incorporate all source data with relatively uniform importance, leading to severe negative transfer. This explains ROOT's superior robustness to hyperparameter variations observed in Section 5.3: ROOT filters data *intrinsically*, while OTDF relies on *extrinsic* manual tuning.

