# OpenReview forum: "Transport or Discard: Robust Unbalanced Optimal Transport for Cross-Domain Policy Adaptation"
_ICML.cc/2026/Conference — ICML 2026 regular_

### Official Review · Reviewer_YNY9 · 2026-02-16

**Soundness:** 3
**Presentation:** 3
**Significance:** 3
**Originality:** 3
**Overall Recommendation:** 4
**Confidence:** 4

**Summary:**

summary:
	In this paper, the authors proposed a novel soft subset selection mechanism to deal with the problem of correlation rankings in traditional data filtering in cross-domain offline RL, termed as ROOT. Empirical and theoretical evidences are provided to demonstrate the effectiveness of the proposed ROOT.

**Compliance With Llm Reviewing Policy:**

Affirmed.

**Final Justification:**

Please refer to my review and comments.

**Key Questions For Authors:**

Please refer to the weakness.

**Limitations:**

No, please carefully answer the questions and issues raised in the weakness.

**Strengths And Weaknesses:**

strength:

	1. The motivation of this paper is clear.

	2. The method proposed is simple yet effective.

	3. Brief theoretical evidences and extensive experimental evidences are provided to show the effectiveness of the proposed ROOT.

	4. The paper is easy to follow.

weakness:

	1. One primary claim of the proposed ROOT is its independence from manual filtering thresholds. However, Proposition 3.2 reveals that the relevance weight $w_i^*$ is exponentially regulated by the temperature factor $\lambda_{src}$. From a functional perspective, could $\lambda_{src}$ be interpreted as a "soft" threshold that implicitly defines the sharp decay of outlier influence? A discussion on the conceptual distinction between this temperature-based suppression and traditional hard-thresholding would be beneficial.

	2. The scale of the constant $B_{tar}$ lacks detailed discussion, which is critical for the tightness of the theoretical bound. If $B_{tar}$ is large (though finite)—specifically, if $B_{tar} \gg C_{min}(x_i)$—the bound in Proposition 3.2 becomes significantly loose, potentially undermining the guaranteed outlier suppression. Elaborating on the factors determining the magnitude of $B_{tar}$ (e.g., target domain compactness or marginal penalties) would strengthen the theoretical foundation.

	3. Certain terminologies require clarification to avoid confusion. In Section 5.1, while the text mentions gravity, friction, and morphology shifts , Table 1 labels these under "Kinematic Shift", which may be misleading. Additionally, in Table 1, the "Source" and "Target" columns use similar naming conventions (e.g., ANT-M vs. MEDIUM) ; explicitly labeling the dataset quality (e.g., Source Dataset Quality vs. Target Dataset Quality) would improve readability.

	4. Consistency in mathematical notation should be maintained. In Eq. 4, the cost matrix is denoted by uppercase $C$, whereas Eq. 3 and its surrounding text use lowercase $c$ to define the ground cost metric. Please clarify if $C$ in Eq. 4 represents the matrix form of the metric $c$ defined earlier.

	5. To further enhance the paper's depth, it would be insightful to provide more theoretical evidence regarding the baseline performance degradation. Specifically, quantifying how much the performance of a source-only policy (trained on $\mathcal{D}_{src}$) drops when evaluated directly in the target domain would establish a clearer "dynamics gap" baseline. This empirical gap would highlight the necessity and efficacy of the "transport-or-discard" mechanism in mitigating negative transfer.

---

> ### Author Rebuttal · Authors · 2026-03-30
>
> We sincerely thank the reviewer for the constructive and insightful feedback. Below, we address your concerns point by point.
>
> ### **Weakness 1: Further discussion on $\lambda_{src}$**
>
> As you insightfully noted, $\lambda_{src}$ functionally acts as a soft threshold. However, the fundamental distinction lies in its mechanism: hard threshold used in previous work is a forced post-processing truncation, which cuts off potentially valuable samples, whereas the temperature factor $\lambda_{src}$ participates in a global continuous optimization process. This enables the weight of each sample to decay smoothly based on its minimum transport cost $C_{\min}(x_i)$. Please see our response to reviewers FLXF and ki7H (Question 4) for a more detailed discussion.
>
> ### **Weakness 2: Further analysis on the scale of constant $B_{tar}$**
>
> We completely agree that the tightness of Proposition 3.2 depends on $B_{tar}$. To clarify this dependency, we derive an explicit upper bound for $B_{tar}$ based on the target-side dual potential:
>
> **Corollary 1 (Explicit Bound for $B_{tar}$):** Let $m_j^* := (\Gamma^{\top}\mathbf{1}) _ j$ denote the optimal transported target marginal, with $\hat{\nu} _ {tar,\min} := \min _ j \hat{\nu} _ {tar,j}$ and $\hat{\nu} _ {tar,\max} := \max _ j \hat{\nu} _ {tar,j}$. Under the same regularity conditions as App. C, and assuming the optimal target marginals are strictly positive and bounded ($0 < m _ {\min} \le m _ j^* \le m _ {\max} < \infty$), the target dual potential satisfies:
> $$|g_j^{(0)}| \le B_{tar}:= \max\{\lambda _ {tar}\log \frac{m _ {\max}}{\hat{\nu} _ {tar,\min}}, \lambda _ {tar}\log \frac{\hat{\nu} _ {tar,\max}}{m _ {\min}}\}$$
>
> This result provides an explicit scale for $B_{tar}$, revealing that it is not an arbitrary black-box constant. **It is jointly determined by $\lambda_{tar}$ and the discrepancy between the transported and empirical target marginals**. Consequently, a larger $\lambda_{tar}$ or a highly uneven target marginal increases $B_{tar}$, which loosens the exponential bound in Prop 3.2. We will add this complementary result to the revision.
>
> ### **Weakness 3 & 4: Clarification on terminology and notation**
>
> We will standardize all terminologies, including explicitly separating dataset qualities from shift types in Table 1, to ensure complete consistency. Regarding the mathematical notation: lowercase $c(x,y)$ represents the ground cost metric defined on the transition space, while uppercase $C$ in Eq. 4 denotes the discrete cost matrix used in the empirical Kantorovich formulation, where its elements are $C_{ij} := c(x_i, y_j)$. We will explicitly state this definition in the revision.
>
> ### **Weakness 5: Theoretical evidence on baseline performance degradation**
>
> While the performance drop of a source-only policy perfectly serves as an empirical diagnosis for the dynamics gap, ROOT's necessity relies on mismatch-aware problem modeling and existing empirical evidence rather than a new theoretical degradation bound:
> 1. **Problem Formulation:** ROOT is motivated by the fact that many source transitions are kinematically infeasible in the target domain, inducing negative transfer. Thus, **ROOT explicitly avoids forcing all source mass to be matched; instead, it continuously suppresses or discards incompatible ones**. The necessity of the transport-or-discard mechanism originates from addressing this problem setting.
> 2. **Empirical Diagnostic vs. Theoretical Prerequisite:** We agree that a source-only vs. target evaluation provides an excellent empirical dynamics-gap baseline. However, proving this theoretically is not a prerequisite for ROOT. **Our core focus is how to safely transfer given a mismatch, rather than theoretically proving whether source-only degrades**. As suggested, we will include the pure empirical performance of source-only policies across all tasks in the appendix as a quantitative diagnostic baseline for the dynamics gap.
> 3. **Existing Gap Evidence:** Our evaluated baselines utilize source information. **ROOT outperforms them, demonstrating that naively incorporating source data is detrimental under large dynamics shifts**. This inherently proves the dynamics-gap effect and supports our main conclusion: selective suppression of incompatible source transitions is crucial for robust transfer.
>
> We hope these clarifications and the new explicit bound address your concerns. We would be grateful if you could consider them in your final assessment."

---

> > ### Author Rebuttal · Reviewer_YNY9 · 2026-04-01
> >
> > Thanks for the authors' feedback. I maintain my original positive scores.

---

> > > ### Author Response · Authors · 2026-04-01
> > >
> > > Thank you for the positive feedback and for taking the time to review our work. We sincerely appreciate your support!

---

### Official Review · Reviewer_tyoG · 2026-03-13

**Soundness:** 2
**Presentation:** 3
**Significance:** 2
**Originality:** 2
**Overall Recommendation:** 4
**Confidence:** 3

**Summary:**

This paper studied the cross-domain adaptation problem. They propose a relaxed version of the optimal transport algorithm that can discard high-cost data and also relax the target data distribution for the optimal transport. Then, for the transported data, they conduct the weighted IQL training.

 They further conduct experiments on the Mujoco tasks.

**Compliance With Llm Reviewing Policy:**

Affirmed.

**Final Justification:**

The rebuttal addresses most of my concerns, and the additional results on some open questions are very interesting to me. So I raise my score to 4.

**Key Questions For Authors:**

1. Why do you still need to regularize the actor? As you already train the Q with target + optimal transport data (not source and should be kinematic feasible in src). And IQL should also have the ability to regularize the policy.

2. see questions in the weakness.

**Limitations:**

See weakness.

**Strengths And Weaknesses:**

The paper is easy to understand and intuitive. The motivation is clear to me.

However, some parts of the algorithm are not well justified. For example, why can you relax the target distribution by expanding it to the near space? Did you validate it? Also, no theoretical analysis is provided, for example, how the transport affects the result, and is there any pattern in the performance of the optimal transport? Further, this paper looks more like an extension of the previous optimal transport paper for the off-dynamics RL. And the experiment results are not very convincing to me as well.

---

> ### Author Rebuttal · Authors · 2026-03-30
>
> Thank you for the constructive and insightful feedback.
>
> ### **Problem 1: Actor regularization**
>
> We emphasize that IQL training cannot guarantee actions fall within the target's feasible region.
> 1. **Critic vs. Actor:** ROOT's weighted critic mitigates the impact of mismatched source data on Q-values, but cannot constrain the actor's generative distribution. The actor may still exploit high-value actions unsupported by actual target data.
> 2. **IQL's implicit conservatism $\neq$ target-feasibility:** IQL stabilizes learning but doesn't ensure physical execution feasibility under dynamics shifts. Our target-only CVAE restricts the actor to the target-supported manifold.
> 3. **Empirical evidence:** The ablation study below shows a performance drop without the CVAE regularization. We will highlight this in the revision.
> |Env|w/o CVAE|ROOT|
> |---|---|---|
> |hop-grav-0.5|43.56±2.43|**49.03±8.18**|
> |hop-fric-0.5|44.79±3.48|**48.39±10.78**|
>
> ### **Weakness 1: Further justification for relaxing the target distribution**
>
> We agree that this point deserves clearer justification. Our intention is not to arbitrarily expand the target distribution into a nearby region. Instead, **ROOT introduces a controlled relaxation to account for empirical uncertainty under limited target samples**.
> 1. **Relaxing empirical constraints, not true distribution:** With limited target data, the empirical distribution consists of sparse discrete atoms. Strict marginal constraints force the transport plan to overfit these atoms, rather than underlying target manifold. The relaxation allows a controlled deviation to mitigate this.
> 2. **Rigorous formulation:** Section 3.2 models this via a KL uncertainty set around the empirical target distribution, equivalent to a KL regularization on the target marginal. It mitigates noise amplification from target sparsity without letting outliers dominate.
> 3. **Ablation stability:** Section 5.3 shows stable performance across a wide parameter range, proving this is a solid design, not a fragile heuristic. To avoid possible misunderstanding, we will revise “relaxing the target distribution” to “modeling uncertainty in the empirical target distribution” in the final version.
>
> ### **Weakness 2: Theoretical analysis on how transport affects the result**
>
> We appreciate this question and have further clarified the theoretical mechanism behind ROOT:
> 1. **Original Theory (link to performance):** Unlike black-box matching, solving the UOT problem (Eq. 9) yields relevance weights for the downstream critic updates (Eq. 10). Proposition 3.2 shows that these weights decay exponentially as the minimum transport cost $C_{min}$ increases, thereby reducing the influence of less transferable data.
> 2. **Expanded Analysis (explicit dependence):**  To more directly address the requested “pattern,” we extend Proposition 3.2 by deriving an explicit bound for the target dual potential (Corollary 1; also discussed in our response to Reviewer YNY9). This analysis shows that the suppression strength is jointly determined by target relaxation $\lambda_{tar}$ and the geometric cost $C_{min}$.
> 3. **Empirical Pattern:** This extended analysis is consistent with our empirical observations: as the domain gap becomes larger, the benefit of ROOT becomes more pronounced. Under severe support mismatch, ROOT tends to downweight or discard physically incompatible transitions, leading to more robust transfer.
>
> We will include this analysis in the final version.
>
> ### **Weakness 3: Difference between ROOT and OTDF**
>
> ROOT is fundamentally different from OTDF, rather than a simple extension, mainly in how it models cross-domain source availability
>
> 1. **Different objective**: OTDF uses balanced OT to rank source data and keeps a fixed ratio. ROOT instead asks **what is genuinely transferable when source–target overlap is uncertain**, by deciding whether source mass should be transported or dropped.
> 2. **Different formulation**: OTDF is a filtering step built on top of OT. ROOT modifies the transport objective itself: with target-side robust relaxation and source-side unbalanced transport, incompatible source mass can be softly discarded rather than forcibly matched.
> 3. **Different theoretical emphasis**: OTDF justifies filtering through OT deviation scores. ROOT analyzes when balanced transport may fail under mismatch (quota effect), and shows that **the learned relevance weights suppress outliers exponentially (Prop. 3.2)**, reducing reliance on manually chosen quantiles.
> 4. **Empirical difference**: Under stronger structural mismatch, OTDF is more affected by forced balanced matching, while ROOT remains more stable by downweighting or discarding incompatible transitions. ROOT also avoids task-specific threshold tuning. **We have uploaded our code to support reproducibility**.
>
> We hope that our clarifications meaningfully address your concerns. We would be grateful if the reviewer could take them into account when forming the final assessment.

---

> > ### Author Rebuttal · Reviewer_tyoG · 2026-04-03
> >
> > Thanks for the response.
> >
> > For weakness 1, I have some further questions (minor), which I believe could be very interesting to think about.
> >
> > As far as I know, some of the other papers have some visualization of the transition using t-SNE. I notice that, for some env-tasks, the target data points are very, very tightly clustered, but for some of the tasks, the target data points are more spread evenly out. How does this pattern affect your "relaxing the target distribution"? Are there parameters that you can choose, and for those tightly clustered data, maybe the relaxation is less?

---

> > > ### Author Response · Authors · 2026-04-05
> > >
> > > We sincerely thank the reviewer for this insightful follow-up. We entirely agree with your intuition. To provide a rigorous, quantitative analysis without relying on t-SNE visualizations (which cannot be uploaded here), we analyzed the halfcheetah-gravity setting, comparing medium and expert target datasets (both containing 5000 transitions). We mainly evaluated **the inherent geometry of the datasets** and **how ROOT’s target marginal allocation responds to different relaxation strengths**.
> > >
> > > ### **1. Quantifying target dataset geometry**
> > > We computed three geometric statistics in the normalized feature space to characterize the clustering patterns:
> > > - **Mean kNN distance**: smaller means tighter local clustering;
> > > - **Trace of covariance**: measures global spread;
> > > - **CV of kNN distance**: larger values indicate stronger local density heterogeneity.
> > >
> > > | target type | mean kNN distance | trace cov | CV of kNN distance |
> > > |---|---:|---:|---:|
> > > | medium | 1.18 | 17.92 | 0.40 |
> > > | expert | 1.02 | 24.67 | 0.73 |
> > >
> > > These statistics confirm your hypothesis: the expert data exhibits stronger local concentration (smaller mean kNN distance) and highly distinct dense clusters (larger Coefficient of Variation of kNN). Conversely, the medium data has a larger mean kNN distance and smaller CV, confirming it is more evenly spread out.
> > >
> > > ### **2. How target geometry dictates required relaxation**
> > >
> > > In our unbalanced OT formulation, $\lambda_{tar}$ acts as a penalty against deviating from the uniform empirical target distribution. Therefore, a smaller $\lambda_{tar}$ provides stronger relaxation, allowing the transported mass to concentrate on fewer target points rather than being forced to match all $N=5000$ points evenly.
> > >
> > > We analyzed the optimized target marginal under varying $\lambda_{tar}$ using the **effective support size**, which measures how many target points actively receive mass and **normalized entropy**, which measures the smoothness of allocation :
> > >
> > > **(1) Expert level result**
> > > | $\lambda_{tar}$ | normalized entropy | effective support size |
> > > |---:|---:|---:|
> > > | 0.05 | 8.21 | 2249.61 |
> > > | 0.10 | 8.44 | 4120.04 |
> > > | 0.50 | 8.51 | 4968.53 |
> > > | 1.00 | 8.52 | 4991.60 |
> > > | 2.00 | 8.52 | 4997.18 |
> > >
> > > **(2) Medium level result**
> > >
> > > | $\lambda_{tar}$ | normalized entropy | effective support size |
> > > |---:|---:|---:|
> > > | 0.05 | 8.30 | 3447.52 |
> > > | 0.10 | 8.43 | 4297.51 |
> > > | 0.50 | 8.51 | 4945.42 |
> > > | 1.00 | 8.52 | 4984.27 |
> > > | 2.00 | 8.52 | 4995.14 |
> > >
> > > Above result mainly shows that:
> > > 1. **Spread-out data utilizes relaxation more broadly**: Under strong relaxation ($\lambda_{tar}=0.05$), the ,edium dataset spreads mass over 3447 points, whereas the tightly clustered expert dataset naturally collapses its mass onto only 2250 points. This proves that dispersed data requires the relaxation to bridge geometric gaps, while tight data uses the relaxation to focus strictly on its dense clusters.
> > > 2. **Tight data requires less relaxation to stabilize**: For both datasets, the allocation stabilizes (approaching the full 5000 support) once $\lambda_{tar} \ge 0.5$.
> > >
> > > ### **3. Conclusion**
> > >
> > > Your intuition is correct. Tightly clustered data naturally concentrates mass, meaning **a weaker relaxation (larger $\lambda_{tar}$) is sufficient to safely map source data** without forcing it into empty space. Spread-out data actively **utilizes stronger relaxation (smaller $\lambda_{tar}$)** to interpolate across its broader, sparser manifold.
> > >
> > > Crucially, our analysis shows that once $\lambda_{tar}$ reaches a moderate regime ($\ge 0.5$), the allocation stabilizes for both geometries. This explains why our default choice of $\lambda_{tar}=0.5$ is empirically robust across diverse tasks, acting as a safe baseline that prevents over-concentration while avoiding strict empirical overfitting. We will also include the corresponding t-SNE visualizations and a more detailed discussion in the revised version.
> > >
> > > We hope these clarifications and the new explicit bound address your concerns. We would be grateful if you could consider them in your final assessment."

---

### Official Review · Reviewer_ki7H · 2026-03-13

**Soundness:** 3
**Presentation:** 3
**Significance:** 3
**Originality:** 3
**Overall Recommendation:** 4
**Confidence:** 3

**Summary:**

This paper proposes a novel data selection method for solving the cross-domain offline reinforcement learning problem. Precvious method such as the OTDF can not adaptively set the ratio leading to inferior performance. This paper proposes Robust Offline unbalanced Optimal Transport (ROOT) to adaptively set the filter ratio. Empirical results show the improved performance of the proposed method.

**Compliance With Llm Reviewing Policy:**

Affirmed.

**Final Justification:**

The rebuttal has addressed most of my concerns. I keep my positive score.

**Key Questions For Authors:**

- Can you provide the discussion on these methods and add the necessary baselines in your experiments?


- Can you provide more experiments on other tasks such as the Antmaze and Adorit to show the generalizability of the method?


- Can you add time and memory analysis compared with the OTDF?

- Can you explain more about the introduction of new hyperparameters $\lambda$ for source and target in equation 9? It seems that you introduce two more paramters need to be tuned than the filter ratio in the previous OTDF.

**Limitations:**

yes

**Strengths And Weaknesses:**

**Strengths**

- This paper proposes a source dataset filtering method that uses the optimal transport to filter out the data close to the target dataset. This paper introduces the unbalanced optimal transport problem and derives an optimization solution for it. The contribution of this paper is nontrivial.

- Theoretical guarantees motivate the design of an adaptive weight method for OTDF. This method has the theoretical support for the claim that using the unbalanced optimal transport to solve the cross-domain offline RL problem.


- It also provides the engineer design for larger-scale cross-domain problems.


- The organization of this paper is clear.


**Weeknesses**

- This paper lacks discussion and comparison with several prior methods [1-4]. Also, this paper misses several cross-domain baselines in the experiments including REAG[3] and Mobody[4].

- The experiments on the MuJoCo control tasks show the strong performance of the proposed methods. It would be better to provide more tasks to show the generalizability of the method. Antmaze and Adorit tasks can be good environments to evaluate.

- Since this method provides optimizations in time and memory. It would be better to incorporate time and memory analysis compared with OTDF.

[1] Cross-domain policy adaptation via value-guided data filtering. Neurips 2023.

[2] Composite Flow Matching for Reinforcement Learning with Shifted-Dynamics Data. Neurips 2025.

[3] Return Augmented Decision Transformer for Off-Dynamics Reinforcement Learning. TMLR 2026.

[4] MOBODY: Model Based Off-Dynamics Offline Reinforcement Learning. ICLR 2026.

---

> ### Author Rebuttal · Authors · 2026-03-30
>
> We sincerely thank the reviewer for the constructive and insightful feedback.
>
> ### **Q1 & W1: Additional baselines**
>
> We have added experimental comparisons with **REAG**[1] and **MOBODY**[2] to the paper, and further clarified the relationship and differences between ROOT and methods [1-4] you mentioned in related work.
>
> |Env|Shift|REAG|MOBODY|ROOT|
> |---|---|---|---|---|
> |Half-Gra|0.1|16.14±0.66|14.18±1.06|**16.33±1.14**|
> |Half-Gra|0.5|40.50±1.58|**47.18±1.23**|39.53±11.36|
> |Half-Gra|2.0|33.28±3.16|41.60±7.35|**43.57±12.73**|
> |Half-Fri|0.1|9.74±0.46|**57.53±2.49**|55.35±5.40|
> |Half-Fri|0.5|66.50±0.99|**69.54±0.48**|67.01±1.13|
> |Half-Fri|2.0|37.74±2.35|**50.02±3.26**|47.21±0.76|
> |Ant-Gra|0.1|15.75±1.17|**37.09±2.12**|32.83±2.91|
> |Ant-Gra|0.5|13.25±0.86|**37.44±2.79**|29.79±6.25|
> |Ant-Gra|2.0|43.25±1.72|45.83±1.71|**58.23±12.14**|
> |Ant-Fri|0.1|54.13±0.56|**58.79±0.11**|55.01±0.14|
> |Ant-Fri|0.5|57.46±0.65|**62.41±4.10**|59.92±0.02|
> |Ant-Fri|2.0|21.28±0.72|47.41±4.40|**55.96±10.49**|
> |Hop-Gra|0.1|31.11±1.80|36.25±1.50|**48.21±9.87**|
> |Hop-Gra|0.5|36.37±2.06|33.57±6.71|**49.03±8.18**|
> |Hop-Gra|2.0|16.44±1.60|23.79±2.09|**27.21±2.53**|
> |Hop-Fri|0.1|33.08±2.53|**51.19±2.56**|51.15±12.47|
> |Hop-Fri|0.5|38.10±3.32|41.34±0.49|**48.39±10.78**|
> |Hop-Fri|2.0|10.20±0.30|11.00±0.14|**11.05±0.04**|
> |Wal-Gra|0.1|26.56±2.62|65.85±5.08|**75.48±2.95**|
> |Wal-Gra|0.5|**55.20±2.18**|43.57±2.32|51.79±5.42|
> |Wal-Gra|2.0|13.50±2.38|44.32±4.58|**61.52±3.97**|
> |Wal-Fri|0.1|10.58±0.71|28.23±9.13|**48.11±6.44**|
> |Wal-Fri|0.5|**78.58±1.08**|76.96±1.99|77.20±10.08|
> |Wal-Fri|2.0|42.18±3.85|73.74±0.49|**73.85±0.47**|
>
> As shown above, ROOT achieves competitive performance across most scenarios compared with the model-based method MOBODY . We will also include more relevant experimental results in the revised paper.
>
> **references**:
>
> [1] Return Augmented Decision Transformer for Off-Dynamics Reinforcement Learning. TMLR 2026.
>
> [2] MOBODY: Model Based Off-Dynamics Offline Reinforcement Learning. ICLR 2026.
>
> ### **Q2 & W2: Evaluation on Adroit/AntMaze tasks**
>
> To further validate generalizability, we evaluated ROOT on the Adroit domain. As shown below, ROOT achieves substantial performance gains over current SOTA methods. AntMaze results will also be included in the revision.
>
> |Env|Type|Level|REAG|MOBODY|OTDF|ROOT|
> |---|---|---|---|---|---|---|
> |Pen|Kin-broken|Med|**62.36±2.60**|37.67±4.54|60.73±4.07|61.87±8.78|
> |Pen|Kin-broken|Hard|26.30±6.49|13.73±6.32|**69.43±6.66**|57.70±5.33|
> |Pen|Morph-shrink|Med|18.06±3.52|16.48±10.46|40.15±15.23|**44.63±6.63**|
> |Pen|Morph-shrink|Hard|18.38±4.59|37.80±1.18|42.64±1.81|**54.40±11.53**|
> |Door|Kin-broken|Med|38.86±7.63|39.26±3.72|58.88±4.27|**64.88±5.57**|
> |Door|Kin-broken|Hard|60.22±5.23|61.61±9.84|70.23±3.02|**71.00±5.40**|
> |Door|Morph-shrink|Med|61.45±0.68|**63.67±9.52**|61.76±5.75|60.35±3.24|
> |Door|Morph-shrink|Hard|62.66±1.60|62.88±5.25|70.22±6.30|**81.02±3.69**|
>
> ### **Q3 & W3: Time and memory analysis vs. OTDF**
>
> |Method|Pre-train Mem|Pre-train Time|Train Mem|Train Time|
> |---|---|---|---|---|
> |OTDF|44290MiB|26.99s|1714MiB|2412s/1M step|
> |ROOT|31356MiB|75.85s|1986MiB|2524s/1M step|
>
> ROOT consumes less pre-training memory because our transport-or-discard mechanism only utilizes the source marginal to derive weights, avoiding the memory-intensive instantiation of the full dense transport matrix. The slightly longer pre-training time is exclusively because ROOT strictly enforces a much larger maximum Sinkhorn iteration limit (1000 vs. OTDF's 100) to guarantee rigorous convergence for the soft selection. This minor one-time tradeoff yields significantly more stable correlation estimates and theoretical consistency.
>
> ### **Q4: Introduction of new hyperparameters ($\lambda_{src}$ and $\lambda_{tar}$)**
>
> These parameters do not replace OTDF's filter ratio; they transform hard-filtering into an adaptive continuous optimization, effectively reducing tuning difficulty:
> 1. **$\lambda_{src}$:** OTDF's filter ratio strictly dictates a hard retention quota. $\lambda_{src}$ instead controls the exponential decay of relevance weights w.r.t minimum transport cost $C_{min}$. It allows geometric compatibility, rather than a fixed cutoff, to adaptively dictate "whether to transport or discard."
> 2. **$\lambda_{tar}$:** Strictly matching sparse empirical target data causes severe overfitting. $\lambda_{tar}$ introduces robust target-side relaxation, allowing controlled deviations around the empirical distribution to mitigate target sparsity noise.
> 3. **Eliminating Task-Specific Tuning:** Sec 5.3 demonstrates both parameters have wide, stable effective ranges. Unlike hard thresholds that are highly task-sensitive, **we strictly use a completely fixed set of parameters across all benchmark tasks**, effectively eliminating the need for manual fine-tuning.
>
> Thank you again for your review and help in improving our paper. Please let us know if you have any additional questions or concerns regarding our paper!

---

> > ### Author Rebuttal · Reviewer_ki7H · 2026-04-03
> >
> > Thanks for the authors' detailed discussion and additional experiments, which have addressed most of my concerns. I hope the authors will incorporate these discussions into the revised manuscript. I keep my postive scores.

---

> > > ### Author Response · Authors · 2026-04-07
> > >
> > > Thank you for the positive feedback and for taking the time to review our work. We sincerely appreciate your support!

---

### Official Review · Reviewer_FLXF · 2026-03-14

**Soundness:** 4
**Presentation:** 3
**Significance:** 4
**Originality:** 3
**Overall Recommendation:** 4
**Confidence:** 3

**Summary:**

The paper provides the alternative strategy of offline reinforcement learning adopted to sort out the challenging issue of the usage of a source dataset to improve policy learning in a data-scarce target domain, which usually happened in cross-domain transport. It is the dramatic framework of robust offline unbalanced optimal transport (ROOT) to build the principled transport-or-discard mechanism. What the most valuable is the ROOT could down-weight or discard high-cost source samples rather than forcing them onto the target support.

**Compliance With Llm Reviewing Policy:**

Affirmed.

**Key Questions For Authors:**

How to calculate the threshold which has to be adopted to decide whether and how many irrelevant transitions are imported to target data?

**Limitations:**

None.

**Strengths And Weaknesses:**

What the most valuable is the ROOT could down-weight or discard high-cost source samples rather than forcing them onto the target support. It builds the strength of paper.
The weakness of paper could be how to calculate the threshold which has to be adopted to decide whether and how many irrelevant transitions are imported to target data.

---

> ### Author Rebuttal · Authors · 2026-03-30
>
> We sincerely thank the reviewer for raising this critical question.
>
> ### **Q & W: How to calculate the threshold**
>
> To clarify, the primary workflow of ROOT can be summarized as:
>
> **solving the Unbalanced Optimal Transport (UOT) problem via the Sinkhorn algorithm (Eq. 9)**
>
> $\rightarrow$ **deriving relevance weights from the source marginal (Eq. 10)**
>
>  $\rightarrow$ **performing weighted Bellman updates (Eq. 12)**
>
> Regarding the concern about thresholding, we provide the following clarifications:
>
> **1. A Paradigm Shift in Filtering:** ROOT does not require a pre-defined hard threshold to explicitly dictate whether or how many source transitions to retain. Instead, **ROOT leverages $\lambda_{src}$ to transform this discrete filtering into a continuous, adaptive transport-or-discard process, essentially answering how strongly should mismatched samples be suppressed**. In previous methods, a cutoff threshold directly dictates the retention ratio (i.e., answering how much to keep). In contrast, $\lambda_{src}$ does not enforce a retention quota, it controls the decay rate of relevance weights as the minimum transport cost $C_{\min}$ increases. Each source transition receives a continuous weight based on its geometric alignment with the target domain (as theoretically bounded in Proposition 3.2): transitions with small $C_{\min}$ maintain high weights, while those with large $C_{\min}$ are automatically penalized and driven near zero, effectively discarding them. This is the fundamental distinction between ROOT and previous methods: **we shift from fixed-quota hard filtering to continuous-weight soft selection**.
>
> **2. The Determination and Transferability of $\lambda_{src}$:** The behavior of $\lambda_{src}$ differs fundamentally from traditional threshold parameters. In our implementation, **we normalize the state-action transitions $(s, a, s')$, ensuring that transport cost scales are strictly aligned across different tasks**. This grants $\lambda_{src}$ exceptional stability and transferability. Furthermore, as demonstrated in the ablation study (Section 5.3), ROOT's performance is highly robust to $\lambda_{src}$ within a broad range, indicating that it does not require meticulous, task-specific fine-tuning. Because of this adaptive nature, we strictly use a single, fixed value of $\lambda_{src} = 0.05$ across all benchmark tasks (see Appendix Table 3). In practice, **users do not need to fine-tune this parameter, which highlights ROOT's significant advantage in generalizability**.
>
> In summary, rather than explicitly calculating a hard threshold to dictate the exact quota of imported source transitions, ROOT leverages the unbalanced optimal transport geometry to achieve adaptive, continuous data selection. This mechanism naturally suppresses irrelevant transitions, **allowing the algorithm to dynamically determine whether and how much source data to utilize based on actual cross-domain alignment, bypassing the need for manual threshold computation**.
>
> We hope this clarifies the conceptual distinction and addresses your concern regarding threshold calculation. We kindly request reconsidering the score to reflect these contributions. Thank you!

---

### Decision · Program_Chairs · 2026-04-30

**Decision:**

Accept (regular)

**Comment:**

This paper presents a data filtering approach for source datasets that leverages optimal transport to identify and retain samples most relevant to the target domain. The authors formulate this as an unbalanced optimal transport problem and develop a corresponding optimization procedure to solve it.

The adaptive weighting strategy for OTDF is grounded in theoretical analysis, providing formal justification for applying unbalanced optimal transport to cross-domain offline reinforcement learning. The theoretical results directly inform and validate the proposed method.

The authors have engaged thoroughly with the review process. They provided comprehensive responses to all feedback and conducted additional experiments to address the raised concerns. All reviewer comments have been satisfactorily resolved.